# MotionBind: Multi-Modal Human Motion Alignment for Retrieval, Recognition, and Generation

**Kaleab A. Kinfu**
University of Pennsylvania
Philadelphia, PA 19104, USA
kinfu@upenn.edu

**René Vidal**
University of Pennsylvania
Philadelphia, PA 19104, USA
vidalr@upenn.edu

## Abstract

Recent advances in multi-modal representation learning have led to unified embedding spaces that align modalities such as images, text, audio, and vision. However, human motion sequences, a modality that is fundamental for understanding dynamic human activities, remains largely unrepresented in these frameworks. Semantic understanding of actions requires multi-modal grounding: text conveys descriptive semantics, vision provides visual context, and audio provides environmental cues. To bridge this gap, we propose MotionBind, a novel architecture that extends the LanguageBind embedding space to incorporate human motion. MotionBind has two major components. The first one is a Multi-Scale Temporal Motion Transformer (MuTMoT) that maps motion sequences to semantically meaningful embeddings. Multimodal alignment is achieved via diverse cross-modal supervision, including motion-text pairs from HumanML3D and KIT-ML, motion-video pairs rendered from AMASS, and motion-video-audio triplets from AIST++. The second component is a Retrieval-Augmented Latent diffusion Model (REALM) that can generate motion sequences conditioned on many modalities. MotionBind achieves state-of-the-art or competitive performance across motion reconstruction, cross-modal retrieval, zero-shot action recognition, and text-to-motion generation benchmarks. The code is available at: https://github.com/vidal-lab/MotionBind.

## 1 Introduction

Recent advances in multi-modal representation learning have led to unified embedding spaces that bridge diverse data modalities. Vision-language models like CLIP [1] and ALIGN [2] first demonstrated that aligning images with natural language enables powerful zero-shot visual recognition capabilities. Building on this idea, more comprehensive models have emerged to simultaneously bind multiple modalities. For example, ImageBind [3] aligns six different modalities (images, video, text, audio, depth, and inertial motion (IMU) data) into a shared embedding space. By training on naturally co-occurring data (*e.g.*, images with captions or videos with audio), ImageBind [3] can relate modalities that were never explicitly paired during training, exhibiting "emergent" cross-modal alignment. Similarly, LanguageBind [4] further generalizes this idea by using language as the central bind to align multiple modalities via contrastive learning, mapping all inputs into a common semantic space. This progress underscores the promise of joint embeddings that connect vision, language, audio, and more to facilitate tasks ranging from cross-modal retrieval to zero-shot recognition.

While prior work [5–8] has explored learning joint representations between human motion and language, human motion remains notably absent from general-purpose multi-modal frameworks. Here, "human motion" refers to the sequence of 3D human poses, which conveys rich information about human movements that are critical for understanding actions, behaviors, and intent. Integrating motion into these multi-modal embedding spaces is vital for grounding action semantics in a broader

39th Conference on Neural Information Processing Systems (NeurIPS 2025).

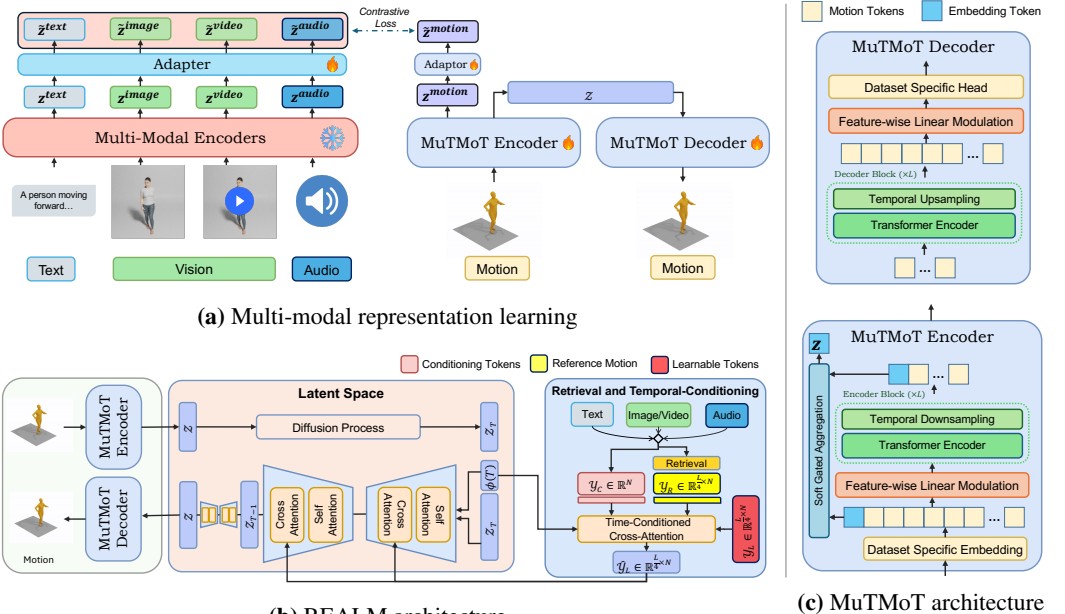

**(a)** Multi-modal representation learning

**(b)** REALM architecture

**(c)** MuTMoT architecture

**Figure 1: Overview of our framework.** (a) We align human motion with text, vision, and audio by projecting modality-specific embeddings (from frozen LanguageBind) and motion embeddings (from MuTMoT) into a shared space via lightweight adapters and a contrastive learning. (b) REALM enables any-to-motion generation through a latent diffusion in the MuTMoT space, using temporal conditioning where learnable frame tokens attend to conditioning input and reference motion via time-aware cross-attention. Together, these components form a unified framework for cross-modal retrieval, recognition, and generation.

context. Effective understanding of human actions requires alignment between motion and other modalities: text conveys descriptions of intent or style, vision (images or video) provides visual cues, and audio can indicate associated sounds or rhythm (*e.g.*, in a dance). Aligning motion with these modalities could enable rich semantic motion representations that would be useful for a more holistic understanding of human motion.

Beyond motion understanding, shared representation spaces have been shown to support "any-to-any" generation, where a single model can generate data like images conditioned on a wide range of modalities [9–11]. Extending this paradigm to motion would allow translating visual cues, musical rhythm, or textual instructions into plausible human motion sequences, opening the door to several applications, including embodied AI and virtual character animation. However, most prior work on motion generation has focused primarily on text-to-motion synthesis [8, 12–15].

In this work, we address these gaps by proposing MotionBind, which aligns human motion with multi-modal embeddings in a unified space for multi-modal motion understanding and synthesis. Building on the LanguageBind framework, we extend its modality set to include human motion, enabling joint reasoning across text, vision, audio, and motion. More specifically, MotionBind makes the following contributions.

1. We introduce a Multi-Scale Temporal Motion Transformer (MuTMoT) encoder-decoder architecture designed to encode human motion sequences into semantically meaningful embeddings aligned with other modalities. The MuTMoT encoder captures motion dynamics at multiple temporal resolutions, producing compact representations that reflect both fine-grained pose transitions and high-level action semantics. The MuTMoT decoder reconstructs motion from the motion latent space, enabling both self-supervised learning and downstream generation.

2. We extend LanguageBind with MuTMoT to unify human motion, text, vision, and audio within a shared embedding space that enables joint cross-modal reasoning and retrieval. The model is trained on a diverse suite of cross-modal datasets, including HumanML3D [16] and KIT-ML [17] for language alignment, AMASS [18] for visual alignment, and AIST++ [19] for joint visual–audio alignment.

3. We present a Retrieval-Augmented Latent Diffusion Model (REALM) that can generate motion sequences conditioned on many modalities by leveraging the multi-modal embeddings of MuTMoT's shared latent space. To improve realism and controllability, REALM retrieves a semantically similar motion from a large database and incorporates it as a reference signal during generation. The generated latent representation is then decoded by MuTMoT to produce the final motion sequence.

4. Extensive experiments on motion reconstruction, cross-modal retrieval, zero-shot action recognition, and text-to-motion synthesis demonstrate that our methods achieve state-of-the-art performance across five benchmarks.

## 2 Related Work

**Multi-Modal Representation Learning.** Recent advances in multi-modal representation learning have led to unified embedding spaces that align diverse modalities through large-scale contrastive training. Early efforts like CLIP [1] and ALIGN [2] established strong image-text alignment by training dual encoders on hundreds of millions of (image, text) pairs, enabling zero-shot recognition and retrieval. Subsequent "foundation" models [20–22] expanded to richer visual inputs, including videos and depth maps. ImageBind [3] extended the paradigm to six modalities (vision, text, audio, depth, thermal, and IMU) by treating images as the central alignment modality. However, this image-centric strategy limits semantic grounding for modalities not naturally paired with vision. LanguageBind [4] addresses this by using a frozen language encoder as the universal anchor and aligning all other modality encoders to language via contrastive learning, yielding improved zero-shot performance on tasks requiring semantic understanding. Despite these advances, human motion remains absent from current multimodal frameworks.

**Motion-Language Representation Learning.** Parallel to the advances in multi-modal models, there has been extensive research on learning joint representations of human motion and language. MotionCLIP [6] introduced a transformer-based motion autoencoder that maps motion sequences into the CLIP embedding space, leveraging both text and image supervision to inherit CLIP's semantic structure and enabling plausible motion generations for text prompts. TM2T [7] proposed a bidirectional framework that tokenizes motion into discrete units and employs a neural machine translation model to facilitate reciprocal generation between text and motion, promoting a shared latent space for both modalities. Building on these foundations, MotionGPT [23] and its successor MotionGPT-2 [8] integrated motion tokens into large language models, creating a unified vocabulary that allows for diverse motion-language tasks, including generation and captioning, within a single framework. Despite these developments, existing models primarily focus on text-motion alignment and often overlook integration with other modalities like audio or vision.

**Motion Retrieval and Action Recognition.** Recent advancements in human motion understanding have emphasized the importance of embedding motion data into semantically rich latent spaces to facilitate tasks such as text-to-motion retrieval and action recognition. Text-to-motion retrieval has evolved from being a mere evaluation metric in text-to-motion synthesis to a primary objective, with models like TMR [5] introducing contrastive learning to align motion and text embeddings effectively. By extending a generative model (TEMOS [13]), TMR demonstrates that combining contrastive and reconstruction losses improves retrieval performance on benchmarks such as HumanML3D and KIT-ML. In parallel, action recognition has benefited from integrating language supervision, as seen in ActionCLIP [24], which models video-text alignment to enable zero-shot recognition without additional labeled data. Despite these advancements, existing approaches often focus on a specific pair of modalities and lack a unified framework that encompasses multiple modalities.

**Text-to-Motion Synthesis.** Text-to-motion synthesis has evolved from early GAN-based approaches, such as Text2Action [25], which generated short and simple actions from natural language, to more expressive and diverse generative models. Variational methods such as T2M [16] and TEMOS [13] modeled probabilistic mappings from text to motion. Autoregressive models [8, 12, 14, 23] introduced discrete motion representations through vector quantization and employed transformer-based sequence models to generate coherent and compositional motion. More recently, diffusion models have emerged as state-of-the-art methods. For instance, MDM [26] and MotionDiffuse [27] adapted denoising diffusion models to motion synthesis, improving generation fidelity. ReMoDiffuse [15] introduced a retrieval-augmented diffusion that incorporates similar motion samples during denoising, and MLD [28] utilized a latent diffusion model to improve efficiency. In our work, we train a

retrieval-augmented latent diffusion model on motion-text pairs but, through alignment with a shared embedding space, we generalize motion synthesis to inputs from text, image, audio, or video.

## 3 Methods

In this work, we present **MotionBind**, which learns a unified embedding space that aligns human motion with other modalities, specifically text, vision, and audio, to enable cross-modal retrieval, action recognition, and motion generation. In Section 3.1, we introduce a motion autoencoder architecture, **MuTMoT**, that encodes motion sequences into this shared space. In Section 3.2, we discuss the multi-modal representation learning paradigm. In Section 3.3, we present **REALM**, a retrieval-augmented latent diffusion model capable of synthesizing realistic motion sequences conditioned on any modality. Together, MuTMoT and REALM form MotionBind, a framework that unifies multi-modal alignment and generation, enabling applications such as cross-modal retrieval, zero-shot action recognition, and any-to-motion synthesis from a shared semantic space. We present the overall architecture of our approach in Figure 1.

### 3.1 Multi-Scale Temporal Motion Transformer

In this section, we present our Multi-Scale Temporal Motion Transformer (MuTMoT), a transformer-based hierarchical architecture designed to encode motion sequences into compact embeddings aligned with a shared multi-modal space, and to decode them back for motion reconstruction or generation. As illustrated in Figure 1c, it consists of a modular encoder-decoder design, supporting both representation learning and generation across several motion benchmarks.

#### 3.1.1 Motion Encoding

**Dataset-Specific Embedding.** Since our approach integrates multiple heterogeneous datasets (AMASS [18], HumanML3D [16], KIT-ML [17], and AIST++ [19]), the number of joints in the pose representations varies across datasets. To handle these discrepancies, we use a dataset-specific embedding module that maps raw motion sequences into a unified representation space. Given a motion sequence $\mathbf{X} \in \mathbb{R}^{T \times C_{ds}}$, where $T$ denotes the sequence length and $C_{ds}$ is the dataset-specific input dimension, we project the motion features to a shared latent space of dimension $D$. Next, we prepend a learnable motion embedding token $\mathbf{z} \in \mathbb{R}^{1 \times D}$ to the motion sequence. To inject temporal order information, we add sinusoidal positional encodings $\mathbf{P} \in \mathbb{R}^{(T+1) \times D}$ to the token sequence.

$$\mathbf{E} = \mathbf{X} + \mathbf{P}, \quad \text{where} \quad \mathbf{X} \leftarrow [\mathbf{z}; \mathbf{X}] \in \mathbb{R}^{(T+1) \times D}. \tag{1}$$

**Feature-wise Linear Modulation (FiLM).** To further accommodate dataset-specific characteristics and improve generalization, we leverage Feature-wise Linear Modulation (FiLM) [29]. FiLM applies an affine transformation to feature activations, conditioning on the dataset source. Formally, given the embedding $\mathbf{E}$, FiLM modulation is applied as:

$$\mathbf{E} \leftarrow \gamma_{ds} \odot \mathbf{E} + \beta_{ds}, \tag{2}$$

where $\gamma_{ds}, \beta_{ds} \in \mathbb{R}^D$ are dataset-specific scaling and bias parameters learned during training, allowing the model to flexibly adapt to dataset-specific distributional differences without disrupting shared representations. The resulting embedding $\mathbf{E}$ is then used as the input to the Encoder blocks.

**Hierarchical Encoding Blocks.** The MuTMoT encoder is a multi-layer transformer architecture designed to hierarchically encode motion sequences across multiple temporal scales, capturing both local and global dependencies. Given the motion embedding $\mathbf{E}$, the encoder processes the sequence through a series of encoder blocks consisting of transformer encoders interleaved with temporal downsampling operations. After each encoder block, all tokens except for the first (the learnable motion embedding token) are passed through a 1D convolutional downsampling layer with kernel size $k = 4$, stride $s = 2$, and padding $p = 1$. This reduces the temporal resolution by a factor of 2.

**Motion Embedding.** At each block $b$, the motion embedding token $\mathbf{E}_0^{(b)} \in \mathbb{R}^{1 \times D}$ is stored, resulting in a stack of $B$ motion embedding tokens across scales. At the end of the final block, these motion embedding tokens are fused using a learnable softmax-weighted average:

$$\mathbf{z}^{\text{motion}} = \sum_{b=1}^{B} \alpha_b \mathbf{E}_0^{(b)} \quad \text{with} \quad \boldsymbol{\alpha} = \text{softmax}(\mathbf{w}) \in \mathbb{R}^B. \tag{3}$$

This fused embedding $\mathbf{z}^{\mathrm{motion}} \in \mathbb{R}^D$ serves as the motion representation in the shared multimodal embedding space.

### 3.1.2 Motion Decoding

**Hierarchical Decoding Blocks.** The MuTMoT decoder is symmetric to the MuTMoT encoder, consisting of Decoder blocks that contain transformer encoders followed by temporal upsampling. Starting from a low-resolution latent sequence (either from the encoder or the latent diffusion model as discussed in Section 3.3), the decoder progressively restores temporal detail at each block.

**Dataset-Specific Head.** After the last block, feature-wise linear modulation is again applied to dataset-specific styles. Finally, the decoded latent sequence is passed through a dataset-specific head, implemented as a lightweight convolutional stack, to map the shared hidden representation back to the original motion format. This modular design ensures generalizability and compatibility with varying motion representations across datasets.

### 3.2 Multi-Modal Representation Learning

Our goal is to learn a unified multi-modal embedding space that semantically aligns human motion data with textual, visual, and audio modalities via contrastive learning. As shown in Figure 1a, we build on the frozen multi-modal encoders from LanguageBind and extend the modality set by utilizing MuTMoT for motion encoding. To enable effective alignment between human motion and other modalities, without forcing motion embeddings to conform directly to the frozen LanguageBind space, which may lack the capacity to fully capture motion-specific semantics, we introduce lightweight adapters on both sides. Specifically, given paired motion sequences $\{\mathbf{X}_i\}_{i=1}^N$ and corresponding non-motion inputs $\{\mathbf{Y}_i\}_{i=1}^N$, where each $\mathbf{Y}_i$ may be text, images, audio or video, we compute motion and non-motion embeddings as:

$$\tilde{\mathbf{z}}_i^{\mathrm{motion}} = f_{\mathrm{MM}}(e_{\mathrm{MM}}(\mathbf{X}_i)), \quad \tilde{\mathbf{z}}_i^{\mathrm{nonmot}} = f_{\mathrm{LB}}(e_{\mathrm{LB}}(\mathbf{Y}_i)), \tag{4}$$

where $e_{\mathrm{MM}}$ and $e_{\mathrm{LB}}$ denote the MuTMoT and LanguageBind encoders, and $f_{\mathrm{MM}}$ and $f_{\mathrm{LB}}$ are lightweight adapters that project embeddings to a common semantic space.

To align the modalities, we use a contrastive InfoNCE loss [30], encouraging embeddings from corresponding (positive) pairs to lie closer in the embedding space while pushing apart non-corresponding (negative) pairs. The contrastive loss $\mathcal{L}_{\mathrm{CL}}$ for a given pair $(\tilde{\mathbf{z}}_i^{\mathrm{motion}}, \tilde{\mathbf{z}}_i^{\mathrm{nonmot}})$ is defined as:

$$\mathcal{L}_{\mathrm{CL}} = -\log \frac{\exp\big(s_{ii}^+/\tau\big)}{\exp\big(s_{ii}^+/\tau\big) + \sum_{j \in \mathcal{N}_i} w_{ij} \exp\big((s_{ij}^- - m_{ij})/\tau\big)}, \tag{5}$$

where $s_{ij}^+$ and $s_{ij}^-$ denote the cosine similarities of positive and negative embedding pairs, respectively; $\tau$ is a temperature hyperparameter, $\mathcal{N}_i$ denotes the set of negative pairs for sample $i$, $w_{ij}$ is a motion-length-aware weighting term, and $m_{ij}$ is a dynamic semantic margin. The weighting factor $w_{ij}$ accounts for differences in motion duration between samples. For a normalized length difference $\Delta_{ij}$, we set

$$w_{ij} = \begin{cases} 1 + \lambda \cdot \Delta_{ij}^2, & \text{if } \Delta_{ij} > \delta \\ 1, & \text{otherwise}, \end{cases} \tag{6}$$

where $\lambda$ and $\delta$ are hyperparameters controlling the strength and threshold of the length-based penalty, respectively. This helps the model better discriminate between motions differing significantly in temporal dynamics, improving the semantic granularity of the embedding space.

In addition to temporal penalties, we introduce a semantic margin $m_{ij}$ to modulate the influence of negative pairs based on the similarity between their associated non-motion inputs. Specifically, the margin is defined as:

$$m_{ij} = \begin{cases} 1 - \cos\big(e_{\mathrm{LB}}(\mathbf{Y}_i), e_{\mathrm{LB}}(\mathbf{Y}_j)\big), & \text{if } \cos\big(e_{\mathrm{LB}}(\mathbf{Y}_i), e_{\mathrm{LB}}(\mathbf{Y}_j)\big) < \rho \\ 0, & \text{otherwise}, \end{cases} \tag{7}$$

where $\rho \in [0, 1]$ is a similarity threshold. If two inputs $\mathbf{Y}_i$ and $\mathbf{Y}_j$ are semantically dissimilar in the LanguageBind embedding space (i.e., below the threshold $\rho$), the corresponding motion-modality pair is penalized more heavily. This encourages the learned motion embeddings to respect the semantic distinctions already encoded in the pre-trained multi-modal space, improving cross-modal alignment quality and retrieval discriminability.

### 3.3 Retrieval-Augmented Latent Diffusion Model for Motion Generation

To generate realistic human motion sequences conditioned on multimodal inputs, we introduce a Retrieval-Augmented Latent Diffusion Model (REALM). Unlike conventional motion diffusion models that directly generate raw motion data [15, 26, 27], REALM operates in the compact latent space defined by the MuTMoT encoder. Specifically, given a conditioning modality (text, video, or audio), REALM synthesizes motion by performing the diffusion process on latent motion representations, effectively leveraging the semantic coherence and expressivity of the MuTMoT embedding space. An overview of the REALM architecture is shown in Figure 1b. In what follows, we describe the overall latent diffusion framework and our retrieval-augmented temporal conditioning module.

#### 3.3.1 Latent Diffusion Framework

Let $\mathbf{X} \in \mathbb{R}^{T \times C_{in}}$ and $\mathbf{z} = e_{\text{MM}}(\mathbf{X}) \in \mathbb{R}^{L \times D}$ denote, respectively, a raw motion sequence and its corresponding latent embedding obtained from the MuTMoT encoder, where $L = T/4$ is the temporally downsampled frame length. We adopt a standard denoising diffusion probabilistic model (DDPM [31]) in this latent space. The forward process corrupts the latent $\mathbf{z}_0 = \mathbf{z}$ over $S$ steps with Gaussian noise:

$$q(\mathbf{z}_s \mid \mathbf{z}_{t-1}) = \mathcal{N}(\mathbf{z}_s; \sqrt{1 - \beta_s}\mathbf{z}_{s-1}, \beta_s\mathbf{I}), \tag{8}$$

$$q(\mathbf{z}_{1:S} \mid \mathbf{z}_0) = \prod_{s=1}^{S} q(\mathbf{z}_s \mid \mathbf{z}_{s-1}), \tag{9}$$

where $\beta_s$ is a fixed noise schedule. The denoising model learns the reverse process:

$$p_\theta(\mathbf{z}_{s-1} \mid \mathbf{z}_s, c) = \mathcal{N}(\mathbf{z}_{s-1}; \mu_\theta(\mathbf{z}_s, c, s), \Sigma_\theta(s)), \tag{10}$$

where $c$ is the fused conditioning signal derived from a multi-modal condition $\mathbf{y}_C$ and a retrieved reference motion $\mathbf{y}_R$ (see Section 3.3.3), and $s$ is the current timestep. We use a parameterized transformer as in [27] to learn $\mu_\theta$ via a noise prediction objective.

#### 3.3.2 Reference Motion Retrieval

To enhance the generation process, we incorporate a retrieval mechanism that selects semantically relevant motion sequences from a database. Given a condition embedding $\tilde{\mathbf{z}}_C \in \mathbb{R}^D$ derived from the multimodal encoder, we first retrieve a set of relevant motion sequences from a large-scale motion database. Retrieval is performed by measuring cosine similarity between the query embedding and the embeddings of candidate motions previously encoded by MuTMoT and stored offline:

$$\tilde{\mathbf{z}}_R = \arg\max_{\tilde{\mathbf{z}}_M \in \mathcal{M}} \text{sim}(\tilde{\mathbf{z}}_C, \tilde{\mathbf{z}}_M), \tag{11}$$

where $\mathcal{M}$ represents the set of candidate motion embeddings, and $\tilde{\mathbf{z}}_R$ denotes the selected reference embeddings. These retrieved reference samples provide semantic and structural context for the diffusion model, enhancing realism and consistency during generation.

#### 3.3.3 Temporal Conditioning with Learnable Frame Tokens

A main contribution of REALM lies in its temporal conditioning module, designed to inject frame-specific and timestep-adaptive conditioning context into the diffusion process. This module introduces a set of learnable latent tokens, one for each frame in the latent motion sequence, that dynamically attend to context throughout the denoising trajectory. Formally, for a latent motion sequence of length $L$, we define a set of $L$ learnable temporal tokens $\mathbf{y}_L \in \mathbb{R}^{L \times D}$, where each token $\mathbf{y}_L^i \in \mathbb{R}^D$ corresponds to the $i$-th frame. These tokens are initialized independently and updated at each denoising step $s$ via a series of cross-attention layers applied to the conditioning context. Let the conditioning input be denoted by $c = [\mathbf{y}_C; \mathbf{y}_R]$, where $\mathbf{y}_C$ is the embedding of the condition modality (*e.g.*, text, video, or audio), and $\mathbf{y}_R$ is the latent representation of a retrieved reference motion. At each timestep $s$, the frame tokens $\mathbf{y}_L^i$ are updated via a Time-Conditioned Cross-Attention (TCC) block, resulting in a temporally contextualized conditioning $\hat{\mathbf{y}}_L \in \mathbb{R}^{L \times D}$.

Each TCC block consists of a cross-attention module that infuses contextual information into the frame tokens, followed by a feedforward network, both modulated by the current timestep embedding

$\phi(s) \in \mathbb{R}^D$. Prior to each transformation, both frame tokens and context tokens are normalized using adaptive layer normalization, which injects the timestep-specific modulation via a learned affine transformation.

$$\text{AdaLN}(h, \phi(s)) = \gamma(s) \cdot \frac{h - \mu(h)}{\sigma(h)} + \beta(s), \quad [\gamma(s), \beta(s)] = \text{MLP}(\phi(s)). \tag{12}$$

The update for frame tokens at timestep $s$ proceeds as follows:

$$\tilde{\mathbf{y}}_L = \text{MHA}\left(\text{AdaLN}(\mathbf{y}_L^{(s)}, \phi(s)), \text{AdaLN}(c, \phi(s)), \text{AdaLN}(c, \phi(s))\right), \tag{13}$$

$$\mathbf{y}'_L = \mathbf{y}_L^{(s)} + \text{Dropout}(\tilde{\mathbf{y}}_L), \tag{14}$$

$$\hat{\mathbf{y}}_L^{(s)} = \mathbf{y}'_L + \text{Dropout}\left(\text{MLP}(\text{AdaLN}(\mathbf{y}'_L, \phi(s)))\right). \tag{15}$$

By stacking $L_{\text{TCC}}$ such blocks, the frame tokens evolve to encode temporally localized semantic cues, adapting their attention to different aspects of the conditioning modality and the retrieved reference motion as the denoising progresses. This temporal condition mechanism allows REALM to dynamically integrate multi-modal semantics at each denoising step, resulting in motion generations that are both contextually grounded and temporally coherent.

### 3.3.4 Training and Inference

As mentioned above, we follow a standard denoising diffusion probabilistic model (DDPM [31]) objective in the MuTMoT latent space, predicting the noise added at each timestep and minimizing the predicted and actual noise. To improve robustness and support flexible conditioning, we adopt the classifier-free guidance technique [32]. During training, we randomly drop 9% of the conditioning embeddings for the input modality and 9% of the retrieved motion reference independently. This encourages the model to learn to generate plausible motion both with and without conditioning input or reference motion. The diffusion process during training uses 1,000 timesteps to allow fine-grained and stable learning. However, at inference time, we accelerate sampling by reducing the number of diffusion steps to 50 using a linear timestep schedule for efficient generation without compromising motion quality. We apply classifier-free guidance by running two parallel forward passes: one with full conditioning and reference motion, and one with null conditioning. We then interpolate the output using a guidance scale that controls the contribution of each output. This encourages the generation to be loyal to the conditioning input and reference motion and improves the fidelity of the generation motion as a result. The final denoised latent representation is decoded by the MuTMoT decoder into a full motion sequence. See Appendix A.2.2 for more details.

## 4 Experiments

**Experimental Setup.** We evaluate our models on four core tasks: motion reconstruction, cross-modal retrieval, zero-shot action recognition, and text-to-motion synthesis. Training is conducted using four diverse human motion datasets: (1) AMASS [18, 33–55], a large-scale motion capture corpus rendered into synthetic video to create motion-video pairs; (2) HumanML3D [16] and (3) KIT-ML [17], both of which provide paired motion and text descriptions covering a wide range of everyday actions; and (4) AIST++ [19], a dance-oriented dataset offering synchronized motion, video, and audio for tri-modal alignment. All four datasets are used to train the MuTMoT encoder-decoder. The REALM generation model is trained specifically on HumanML3D and KIT-ML.

**Motion Reconstruction.** We evaluate reconstruction quality using the Fréchet Inception Distance (FID) and the Mean Per Joint Position Error (MPJPE), which respectively measure the distributional similarity and the per-frame positional accuracy between reconstructed and ground truth motions. As shown in Table 1, MuTMoT achieves the lowest FID and MPJPE on both KIT-ML and HumanML3D test sets, outperforming prior autoencoding approaches such as M2DM, T2M-GPT, and MoMask, thereby validating the effectiveness of our multi-scale transformer architecture.

**Cross-Modal Retrieval.** We assess cross-modal retrieval performance on all datasets using Recall at top-k (R@1, R@2, R@3, R@5) in both directions: modality-to-motion and motion-to-modality. Table 2 shows that MuTMoT outperforms previous models such as TEMOS [13] and TMR [5], achieving higher recall across all ranks. This demonstrates the quality of the shared embedding

**Table 1:** Reconstruction results on KIT-ML and HumanML3D test sets. Lower is better for FID, MPJPE, and ACCL. $\pm$ indicates a 95% confidence interval. **Bold** indicates the best, and underline indicates the second-best.

| Method | KIT-ML | | | HumanML3D | | |
|---|---|---|---|---|---|---|
| | FID $\downarrow$ | MPJPE $\downarrow$ | ACCL $\downarrow$ | FID $\downarrow$ | MPJPE $\downarrow$ | ACCL $\downarrow$ |
| ACTOR [58] | - | - | - | $0.341^{\pm.001}$ | 65.3 | 7.0 |
| MLD-1 [28] | - | - | - | $0.247^{\pm.001}$ | 54.4 | 8.3 |
| T2M-GPT [14] | $0.472^{\pm.011}$ | - | - | $0.070^{\pm.001}$ | 58.0 | - |
| MotionGPT [8] | - | - | - | $0.067^{\pm.001}$ | 55.8 | 7.5 |
| M2DM [59] | $0.413^{\pm.009}$ | - | - | $0.063^{\pm.001}$ | - | - |
| MoMask [12] | $0.112^{\pm.002}$ | 37.2 | - | **$0.019^{\pm.001}$** | 29.5 | - |
| **MuTMoT (Ours)** | **$0.077^{\pm.001}$** | **23.5** | **10.8** | $0.031^{\pm.001}$ | **25.6** | **4.7** |

space in capturing fine-grained semantic correspondence between motion and other modalities. For AMASS, which lacks an official split, we constructed a 70/30 train-test split for evaluation.

**Zero-Shot and Few-Shot Action Recognition.** We evaluate MuTMoT on BABEL-60 and BABEL-120 [56] under zero-shot, few-shot, and supervised settings. In the zero-shot setting, action labels are converted into natural language descriptions using GPT-4 [57] and encoded by the text encoder, and motion samples are encoded by MuTMoT and compared with these text embeddings via cosine similarity. In the few-shot setting ($N = 10$), class prototypes are computed by averaging embeddings of 10 labeled motions per class, and predictions combine few-shot and zero-shot logits as scores = $\xi \cdot \text{FS} + (1 - \xi) \cdot \text{ZS}$ with $\xi = 0.7$. A supervised linear probe further trains a linear classifier on top of the frozen MuTMoT embeddings. In cases where a motion sample has multiple ground-truth labels (as common in BABEL), the prediction is considered correct if any of these labels appears within the Top-1 or Top-5 results. As shown in Table 3, MuTMoT ($N = 0$) yields over 50% Top-5 accuracy in zero-shot classification, while the linear probe matches the performance of a fully supervised state-of-the-art method. This demonstrates robust language-motion alignment even for fine-grained or overlapping categories (*e.g.*, *walk* vs. *forward movement*).

**Text-to-Motion Generation.** We evaluate the generative quality of REALM against state-of-the-art methods using R-Precision, FID, multimodality, and diversity, following HumanML3D and KIT-ML protocols. As it can be seen in Table 4, MuTMoT achieves competitive performance across all metrics, while additionally supporting conditioning from non-text modalities such as audio and video.

We also present qualitative examples that illustrate the semantic fidelity, diversity, and modality flexibility of our model. Figure 2 shows results for the text-to-motion synthesis task. The generated motions accurately capture the fine-grained semantics of input prompts, such as "a person climbs up a ladder" and "a man picks up an object using his left hand", demonstrating smooth transitions and coherent articulation consistent with natural human movement.

Due to page limits, we have included additional results, ablation studies, more qualitative results, and architectural and implementation details, in the Appendix.

Altogether, our experiments demonstrate that the proposed methods not only perform strongly on standard motion-language benchmarks but also extend effectively to broader multi-modal settings, showing generalization in retrieval, recognition, and generation tasks.

## 5 Limitations

While our framework demonstrates strong performance across a range of benchmarks and introduces new capabilities for multi-modal motion generation, several limitations remain. First, although REALM supports any-to-motion generation by conditioning on embeddings from arbitrary modalities (*e.g.*, text, image, video, audio), our quantitative evaluation is currently limited to text-conditioned generation. As such, while qualitative results indicate promising generalization, further work is needed to robustly assess generation quality across all modalities. Second, our model relies on the assumption that modality-specific encoders from LanguageBind provide sufficiently expressive representations. This assumption may not hold in cases where these encoders fail to capture the nuances of the input, potentially limiting the effectiveness of motion retrieval or generation. Third, the performance

**Table 2:** Cross-modal retrieval results on HumanML3D (Text), KIT-ML (Text), AMASS (Video), and AIST++ (Video+Audio) test sets. Recall@k metrics are shown for both input-to-motion and motion-to-input retrieval directions. **Bold** indicates the best, and underline indicates the second-best.

| Dataset | Method | Input-to-Motion Retrieval | | | | | Motion-to-Input Retrieval | | | | |
|---|---|---|---|---|---|---|---|---|---|---|---|
| | | R@1↑ | R@2↑ | R@3↑ | R@5↑ | MedR↓ | R@1↑ | R@2↑ | R@3↑ | R@5↑ | MedR↓ |
| HumanML3D | TEMOS [13] | 40.49 | 53.52 | 61.14 | 70.96 | 2.33 | 39.96 | 53.49 | 61.79 | 72.40 | 2.33 |
| | Guo et al. [16] | 52.48 | 71.05 | 80.65 | 89.66 | 1.39 | 52.00 | 71.21 | 81.11 | 89.87 | 1.38 |
| | TMR [5] | 67.16 | 81.32 | 86.81 | 91.43 | 1.04 | 67.97 | 81.20 | 86.35 | 91.70 | 1.03 |
| | **MuTMoT** | **69.56** | **85.00** | **90.45** | **94.86** | **1.00** | **70.65** | **86.92** | **91.42** | **96.26** | **1.00** |
| KIT-ML | TEMOS [13] | 43.88 | 58.25 | 67.00 | 74.00 | 2.06 | 41.88 | 55.88 | 65.62 | 75.25 | 2.25 |
| | Guo et al. [16] | 42.25 | 62.52 | 75.12 | 87.56 | 1.88 | 39.75 | 62.75 | 73.62 | 86.88 | 1.95 |
| | TMR [5] | 49.25 | 69.75 | 78.25 | 87.88 | 1.50 | 50.12 | 67.12 | 76.88 | 88.88 | 1.53 |
| | **MuTMoT** | **51.36** | **70.44** | **79.56** | **88.83** | **1.43** | **50.59** | **69.62** | **80.25** | **89.37** | **1.47** |
| AMASS | **MuTMoT** | **97.16** | **99.61** | **99.77** | **99.84** | **1.00** | **96.96** | **99.27** | **99.75** | **99.89** | **1.00** |
| AIST++ | **MuTMoT** | **70.00** | **87.50** | **90.00** | **95.00** | **1.00** | **62.50** | **82.50** | **90.00** | **95.00** | **1.15** |

**Table 3:** Top-1 and Top-5 classification accuracy on BABEL-60 and BABEL-120. MuTMoT ($N = 0$) denotes zero-shot classification; MuTMoT ($N = 10$) combines zero-shot alignment with 10-shot support via prototype-based classification. MuTMoT (BABEL) is trained using a linear classifier on frozen embeddings. **Bold** indicates the best, and underline indicates the second-best in the supervised setting.

| Method | Training Data | BABEL-60 | | BABEL-120 | |
|---|---|---|---|---|---|
| | | Top-1 ↑ | Top-5 ↑ | Top-1 ↑ | Top-5 ↑ |
| 2s-AGCN [56, 60] CE | BABEL | 41.14 | **73.18** | 38.41 | 70.49 |
| 2s-AGCN [56, 60] Focal | BABEL | 33.41 | 67.83 | 27.91 | 57.96 |
| MotionCLIP [6] | BABEL | 40.90 | 57.71 | - | - |
| MuTMoT (BABEL) | BABEL | **42.30** | 72.40 | **40.10** | **70.70** |
| MuTMoT ($N = 10$) | HumanML3D | 28.07 | 56.89 | 24.10 | 49.62 |
| MuTMoT ($N = 0$) | HumanML3D | 21.24 | 53.48 | 20.01 | 51.10 |

of REALM is influenced by the quality and coverage of the retrieved reference motions. While retrieval improves realism and control, the absence of semantically relevant references, especially in out-of-distribution scenarios, may lead to suboptimal or generic outputs. Finally, although we train on four diverse motion datasets and evaluate on multiple tasks, broader evaluations on diverse movement types and environmental context remains an important area for future exploration. We believe addressing these limitations will further enhance the generality, robustness, and applicability of multi-modal motion models in future works.

# 6 Conclusion

In this work, we presented MotionBind, a unified framework for integrating human motion into multi-modal representation learning and generation. Our approach extends the LanguageBind embedding space to incorporate human motion alongside text, images, audio and video. Central to our framework is the Multi-Scale Temporal Motion Transformer (MuTMoT), a transformer-based encoder-decoder architecture that captures fine-grained and long-range motion dynamics while enabling alignment with other modalities through lightweight adapters and contrastive learning. To support motion generation in this shared space, we introduced REALM, a retrieval-augmented latent diffusion model that operates in the latent space of MuTMoT. REALM conditions on multi-modal embeddings and semantically retrieved reference motions, and employs a novel temporal conditioning mechanism with learnable frame-wise tokens to generate realistic, context-aware motion sequences. Extensive experiments across motion reconstruction, cross-modal retrieval, zero-shot action recognition, and text-to-motion synthesis demonstrated that our approach achieves state-of-the-art or competitive performance, while simultaneously enabling "emergent" capabilities such as multi-modal retrieval and any-to-motion generation.

**Table 4:** Text-to-motion generation evaluation on the HumanML3D and KIT-ML test sets. $\pm$ indicates a 95% confidence interval. **Bold** indicates the best, and underline indicates the second-best.

| Dataset | Method | R@1 ↑ | R@2 ↑ | R@3 ↑ | FID ↓ | MM Dist ↓ | MModality ↑ |
|---|---|---|---|---|---|---|---|
| HumanML3D | TM2T [7] | $0.424^{\pm.003}$ | $0.618^{\pm.003}$ | $0.729^{\pm.002}$ | $1.501^{\pm.017}$ | $3.467^{\pm.011}$ | $2.424^{\pm.093}$ |
| | T2M [16] | $0.455^{\pm.003}$ | $0.636^{\pm.003}$ | $0.736^{\pm.002}$ | $1.087^{\pm.021}$ | $3.347^{\pm.008}$ | $2.219^{\pm.074}$ |
| | MDM [26] | - | - | $0.611^{\pm.007}$ | $0.544^{\pm.044}$ | $5.566^{\pm.027}$ | $\mathbf{2.799}^{\pm.072}$ |
| | MLD [28] | $0.481^{\pm.003}$ | $0.673^{\pm.003}$ | $0.772^{\pm.002}$ | $0.473^{\pm.013}$ | $3.196^{\pm.010}$ | $2.413^{\pm.079}$ |
| | MotionDiffuse [27] | $0.491^{\pm.001}$ | $0.681^{\pm.001}$ | $0.782^{\pm.001}$ | $0.630^{\pm.001}$ | $3.113^{\pm.001}$ | $1.553^{\pm.042}$ |
| | T2M-GPT [14] | $0.492^{\pm.003}$ | $0.679^{\pm.002}$ | $0.775^{\pm.002}$ | $0.141^{\pm.005}$ | $3.121^{\pm.009}$ | $1.831^{\pm.048}$ |
| | MotionGPT [23] | $0.492^{\pm.003}$ | $0.681^{\pm.003}$ | $0.778^{\pm.002}$ | $0.232^{\pm.008}$ | $3.096^{\pm.008}$ | $2.008^{\pm.084}$ |
| | ReMoDiffuse [15] | $0.510^{\pm.005}$ | $0.698^{\pm.006}$ | $0.795^{\pm.004}$ | $\underline{0.103}^{\pm.006}$ | $\underline{2.974}^{\pm.016}$ | $1.795^{\pm.043}$ |
| | MoMask [12] | $\mathbf{0.521}^{\pm.002}$ | $\mathbf{0.713}^{\pm.002}$ | $\mathbf{0.807}^{\pm.002}$ | $\mathbf{0.045}^{\pm.002}$ | $\mathbf{2.958}^{\pm.008}$ | $1.241^{\pm.040}$ |
| | **REALM (Ours)** | $\underline{0.511}^{\pm.003}$ | $\underline{0.701}^{\pm.002}$ | $\underline{0.801}^{\pm.002}$ | $0.204^{\pm.004}$ | $2.980^{\pm.008}$ | $\underline{2.483}^{\pm.040}$ |
| KIT-ML | TM2T [7] | $0.280^{\pm.005}$ | $0.463^{\pm.006}$ | $0.587^{\pm.005}$ | $3.599^{\pm.153}$ | $4.591^{\pm.026}$ | $\mathbf{3.292}^{\pm.081}$ |
| | T2M [16] | $0.361^{\pm.005}$ | $0.559^{\pm.007}$ | $0.681^{\pm.007}$ | $3.022^{\pm.107}$ | $3.488^{\pm.028}$ | $2.052^{\pm.107}$ |
| | MDM[28] | - | - | $0.396^{\pm.004}$ | $0.497^{\pm.021}$ | $9.191^{\pm.022}$ | $1.907^{\pm.214}$ |
| | MLD [28] | $0.390^{\pm.008}$ | $0.609^{\pm.008}$ | $0.734^{\pm.007}$ | $0.404^{\pm.027}$ | $3.204^{\pm.027}$ | $\underline{2.192}^{\pm.071}$ |
| | MotionDiffuse [27] | $0.417^{\pm.004}$ | $0.621^{\pm.004}$ | $0.739^{\pm.004}$ | $1.954^{\pm.062}$ | $2.958^{\pm.005}$ | $0.730^{\pm.013}$ |
| | T2M-GPT [14] | $0.416^{\pm.006}$ | $0.627^{\pm.006}$ | $0.745^{\pm.055}$ | $0.514^{\pm.026}$ | $3.007^{\pm.012}$ | $1.570^{\pm.039}$ |
| | ReMoDiffuse [15] | $0.427^{\pm.014}$ | $0.641^{\pm.014}$ | $0.765^{\pm.006}$ | $\mathbf{0.155}^{\pm.026}$ | $2.814^{\pm.029}$ | $1.239^{\pm.028}$ |
| | MoMask [12] | $\underline{0.433}^{\pm.007}$ | $\underline{0.656}^{\pm.006}$ | $\underline{0.781}^{\pm.005}$ | $\underline{0.204}^{\pm.011}$ | $\underline{2.779}^{\pm.022}$ | $1.131^{\pm.043}$ |
| | **REALM (Ours)** | $\mathbf{0.435}^{\pm.009}$ | $\mathbf{0.661}^{\pm.010}$ | $\mathbf{0.785}^{\pm.008}$ | $0.264^{\pm.010}$ | $\mathbf{2.718}^{\pm.029}$ | $0.881^{\pm.079}$ |

**Figure 2:** Text-to-motion generation results from six different prompts. The generated motions closely follow the semantics of the input descriptions.

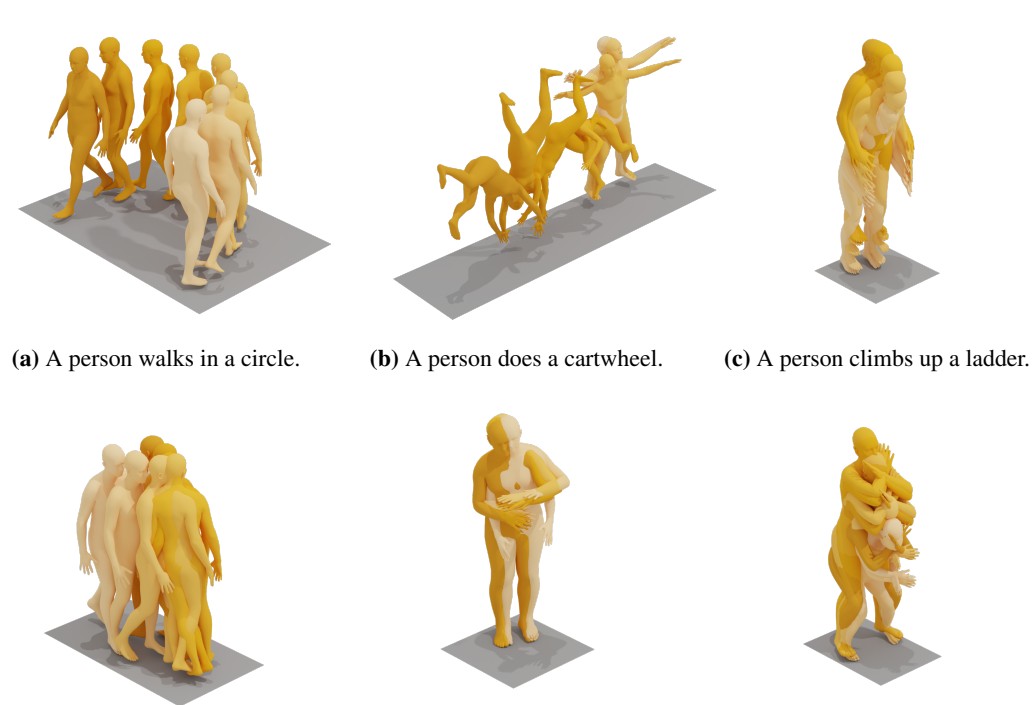

**(a)** A person walks in a circle.

**(b)** A person does a cartwheel.

**(c)** A person climbs up a ladder.

**(d)** Making a left turn.

**(e)** A man picks up an object using left hand, cleans it and puts it back.

**(f)** A person crosses his arms and squats down twice.

## Acknowledgments and Disclosure of Funding

The authors thank Lingjie Liu, Zhiyang (Frank) Dou, Yuyan Ge, and Darshan Thaker for their valuable input and feedback. This research was supported by NSF grant 2124277, NIH grant R01NS135613, and University of Pennsylvania Startup Funds.The views and conclusions expressed in this work are those of the authors and do not necessarily reflect the official policies, either expressed or implied, of NSF, NIH, or the U.S. Government. The U.S. Government is authorized to reproduce and distribute reprints for governmental purposes, notwithstanding any copyright notice herein.

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

# Appendix

# A  Architectural and Implementation Details

In this section, we provide comprehensive details on the architectural and implementation aspects of our proposed methods, Multi-Scale Temporal Motion Transformer (MuTMoT) and Retrieval-Augmented Latent Diffusion Model (REALM).

## A.1  Dataset Preparation and Preprocessing

As outlined in the main paper, we use four publicly available human motion datasets in our experiments: AMASS [18], HumanML3D [16], KIT-ML [17], and AIST++ [19]. These datasets vary in motion diversity, sequence length, joint representation formats, and available modality pairs:

- **AMASS** is a large-scale motion capture dataset aggregated from multiple sources. It is used to generate synthetic motion-video pairs by rendering 3D motion sequences to RGB videos using the SMPL mesh.
- **HumanML3D** and **KIT-ML** provide natural language descriptions paired with 3D human motion sequences, making them well-suited for text-to-motion synthesis and cross-modal retrieval tasks.
- **AIST++** consists of synchronized 3D motion, video, and audio data, primarily focused on dance sequences, enabling tri-modal learning across motion, vision, and music.

To ensure consistency across datasets, all motion data are standardized following the preprocessing pipeline of HumanML3D. In particular, we convert SMPL pose parameters to a unified 22-joint skeleton representation, except for KIT-ML, which uses a 21-joint configuration. The root joint of each sequence is centered at the origin to achieve translation invariance, and the motion is normalized to maintain consistent orientation and scale. Temporal normalization is also applied: sequences longer than 196 frames (at 20 FPS) are segmented into fixed-length clips, while shorter sequences are zero-padded to ensure consistent input dimensionality across the training pipeline.

To augment the textual diversity and improve robustness in cross-modal training, we generate five paraphrases for each description in HumanML3D and KIT-ML using the GPT-4o model. These paraphrases are generated in a consistent style with the original annotations and enable improved generalization in text-to-motion synthesis and retrieval. The prompt used to generate paraphrases is as follows:

```
You are a skilled technical writer trained to generate diverse
and natural paraphrases for human action descriptions.  Given
a sentence describing a human motion, output 5 distinct
alternative phrasings that preserve the exact human action
description but differ in wording and phrasing.  Avoid
redundancy or trivial rewordings.  Make sure each paraphrase
is fluent and consistent with the HumanML3D annotation style.
```

## A.2  Model Architectures

### A.2.1  Multi-Scale Temporal Motion Transformer (MuTMoT)

MuTMoT is a hierarchical transformer-based encoder-decoder designed to represent human motion sequences at multiple temporal scales. This section details its four primary modules: (i) Dataset-specific Embedding, (ii) MuTMoT Encoder, (iii) MuTMoT Decoder, and (iv) Dataset-specific Output Head.

**Dataset-Specific Embedding.**  Given the diversity in input dimensionality across datasets, we standardize input motions into a uniform hidden dimension $D = 512$. Specifically, an input motion sequence $\mathbf{X} \in \mathbb{R}^{T \times C_{ds}}$ undergoes a dataset-specific 1D convolution with kernel size 3, stride 1, and padding 1. Subsequently, a learnable motion embedding token $\mathbf{z} \in \mathbb{R}^{1 \times D}$ is prepended to the projected sequence, resulting in:

$$\mathbf{X} \leftarrow [\mathbf{z}; \mathbf{X}] \in \mathbb{R}^{(T+1) \times D} \quad \text{where} \quad \mathbf{X} \leftarrow \text{Conv1D}(\mathbf{X}). \tag{16}$$

To inject temporal order information, we add sinusoidal positional encodings $\mathbf{P} \in \mathbb{R}^{(T+1) \times D}$ to the token sequence. The resulting embedding is then normalized with LayerNorm (LN) and regularized with dropout rate of 0.1:

$$\mathbf{E} = \text{Dropout}(\text{LN}(\mathbf{X})) \quad \text{where} \quad \mathbf{X} \leftarrow \mathbf{X} + \mathbf{P}. \tag{17}$$

To accommodate dataset-specific distributional differences without compromising shared representational learning, Feature-wise Linear Modulation (FiLM) [29] is applied:

$$\mathbf{E} \leftarrow \gamma_{ds} \odot \mathbf{E} + \beta_{ds}, \tag{18}$$

where $\gamma_{ds}, \beta_{ds} \in \mathbb{R}^D$ are dataset-specific scaling and bias parameters learned during training. The resulting embedding $\mathbf{E}$ is then used as the input to the MuTMoT encoder.

**Motion Encoding.** The MuTMoT encoder comprises $B = 3$ hierarchical transformer blocks, designed to capture both fine-grained and long-range temporal dynamics. Each transformer block includes multi-head self-attention (MHSA) with 4 heads and a feed-forward network (FFN) with an intermediate dimension of 1024, each wrapped by residual connections and LayerNorm and dropout regularization (with dropout rate 0.1): Formally, at layer $b$, the input $\mathbf{E}^{(b)} \in \mathbb{R}^{(T+1) \times D}$ is updated as:

$$\mathbf{E}^{(b+1)} = \text{FFN}\left(\text{LN}_2\left(\mathbf{A}^{(b)}\right)\right) + \mathbf{A}^{(b)} \quad \text{where} \quad \mathbf{A}^{(b)} = \text{MHSA}\left(\text{LN}_1\left(\mathbf{E}^{(b)}\right)\right) + \mathbf{E}^{(b)}, \tag{19}$$

where $\mathbf{E}^{(b)} \in \mathbb{R}^{(T_b+1) \times D}$ denotes embeddings at the b-th layer, and $T_b$ represents the sequence length at scale $b$. After each transformer block, all tokens except for the first (the learnable motion embedding token) are passed through a 1D convolutional downsampling layer with kernel size 4, stride 2, and padding 1. This reduces the temporal resolution by a factor of 2:

$$\mathbf{E}_{1:T_b}^{(b+1)} \leftarrow \text{Conv1D}(\mathbf{E}_{1:T_b}^{(b+1)}). \tag{20}$$

The first token, a learnable motion embedding $\mathbf{E}_0^{(b)} \in \mathbb{R}^{1 \times D}$, is preserved across all blocks to accumulate global sequence information. At the final block, motion embedding tokens from each scale are fused via learned softmax-weighted aggregation:

$$\mathbf{z}_{\text{motion}} = \sum_{b=1}^{B} \alpha_b \mathbf{E}_0^{(b)} \quad \text{where} \quad \boldsymbol{\alpha} = \text{softmax}(\mathbf{w}) \in \mathbb{R}^B, \tag{21}$$

where $\mathbf{w} \in \mathbb{R}^{\mathbf{B}}$ are learnable weights. The resulting compact embedding $\mathbf{z}_{\text{motion}}$ represents the entire motion sequence within the unified multimodal embedding space.

**Motion Decoding.** The decoder reconstructs motion sequences from latent embeddings provided by either the encoder or the latent diffusion model. Mirroring the encoder architecture, the decoder also contains $B = 3$ transformer blocks, each composed of MHSA and FFN sub-blocks with residual and normalization layers:

$$\mathbf{Z}^{(b+1)} = \text{FFN}\left(\text{LN}_2\left(\mathbf{A}^{(b)}\right)\right) + \mathbf{A}^{(b)} \quad \text{where} \quad \mathbf{A}^{(b)} = \text{MHSA}\left(\text{LN}_1\left(\mathbf{Z}^{(b)}\right)\right) + \mathbf{Z}^{(b)}, \tag{22}$$

where $\mathbf{Z}^{(b)} \in \mathbb{R}^{T_b \times D}$. The decoder employs temporal upsampling between layers (except the final layer) using nearest-neighbor interpolation followed by 1D convolution (kernel size 3, padding 1), doubling the temporal resolution progressively at each stage until the original sequence length $T$ is recovered:

$$\widetilde{\mathbf{Z}}^{(l+1)} = \text{Conv1D}\left(\text{Upsample}\left(\mathbf{Z}^{(l+1)}\right)\right). \tag{23}$$

**Dataset-Specific Head.** After the decoder reconstructs the motion embeddings at full temporal resolution, a dataset-specific FiLM layer and a convolutional output head map embed the motion embeddings back to the original dataset-specific dimensions. The output head consists of three convolutional layers, each with kernel size 3, stride 1, and padding 1, formally expressed as:

$$\widetilde{\mathbf{X}} = \text{Conv1D}_{ds}\left(\text{Conv1D}\left(\text{Conv1D}\left(\widetilde{\mathbf{X}}_{ds}\right)\right)\right) \quad \text{and} \quad \widetilde{\mathbf{X}}_{ds} = \gamma'_{ds} \odot \mathbf{Z}^{(L)} + \beta'_{ds}, \tag{24}$$

where $\gamma'_{ds}, \beta'_{ds} \in \mathbf{R}^D$ are learned dataset-specific FiLM parameters. The resulting output $\widetilde{\mathbf{X}}$ reconstructs the motion sequence in the appropriate format for the original dataset.

**Training Objectives.** MuTMoT is trained using a hybrid objective that combines a reconstruction loss and a cross-modal contrastive loss to simultaneously enforce input fidelity and semantic alignment. To ensure that the latent embeddings produced by MuTMoT retain sufficient information to reconstruct the original motion sequence, we employ a dataset-specific reconstruction loss at the output of the decoder. Specifically, we use a Smooth L1 loss (also known as Huber loss []) between the reconstructed motion sequence $\widetilde{\mathbf{X}}$ and the ground truth motion $\mathbf{X}$. This loss is computed independently for each dataset $d \in \mathcal{D}$ based on its skeletal structure, with the joint dimensionality set to 22 for HumanML3D, AMASS, and AIST++, and 21 for KIT-ML. The loss is applied element-wise across the joint dimensions and timesteps, and then averaged over the sequence. Formally, the reconstruction loss is defined as:

$$\mathcal{L}_{recon}^{(d)} = \frac{1}{T \times C_{ds}} \sum_{t=1}^{T} \sum_{j=1}^{C_{ds}} \text{Smooth}_{L1}(\widetilde{\mathbf{X}}_{t,j}, \mathbf{X}_{t,j}). \tag{25}$$

The overall reconstruction loss is the sum across all datasets:

$$\mathcal{L}_{recon} = \sum_{d \in \mathcal{D}} \mathcal{L}_{recon}^{(d)}. \tag{26}$$

To align motion embeddings with those of other modalities (*e.g.*text, audio, video), we adopt a contrastive learning framework based on the InfoNCE loss. This objective encourages paired samples from different modalities to lie close in the shared embedding space, while unpaired samples are pushed apart. Specifically, for each motion input $\mathbf{X}_i$, we compute an embedding $\mathbf{z}_i^{\text{motion}} \in \mathbf{R}^D$ via the MuTMoT encoder $e_{\text{HM}}(\cdot)$, where $D = 512$. This embedding is projected to a higher-dimensional space $\mathbb{R}^{768}$ via a learned linear projection layer:

$$\hat{\mathbf{z}}_i^{\text{motion}} = \mathbf{W}_{\text{proj}} \mathbf{z}_i^{\text{motion}} \tag{27}$$

where $\mathbf{W}_{\text{proj}} \in \mathbf{R}^{768}$. We then apply a lightweight residual-style adapter module to modulate this embedding while preserving expressivity. The adapter is a two-layer MLP with ReLU non-linearity and dropout, followed by a residual connection:

$$\tilde{\mathbf{z}}_i^{\text{motion}} = f_{\text{HM}}(\hat{\mathbf{z}}_i^{\text{motion}}) \tag{28}$$

$$f_{\text{HM}}(\hat{\mathbf{z}}_i^{\text{motion}}) = \hat{\mathbf{z}}_i^{\text{motion}} + \mathbf{W}_2(\text{Dropout}(\text{ReLU}(\mathbf{W}_1 x))), \tag{29}$$

with $\mathbf{W}_1, \mathbf{W}_2 \in \mathbf{R}^{768 \times 768}$ and a dropout probability of 0.1. An identical adapter architecture is used for the frozen LanguageBind modality-specific embeddings, denoted $\tilde{\mathbf{z}}_i^{\text{modality}}$.

We optimize the following contrastive loss over a batch of $N$ motion-modality pairs:

$$\mathcal{L}_{\text{CL}} = -\frac{1}{N} \sum_{i=1}^{N} \log \frac{\exp\left(\frac{s_{ii}^+}{\tau}\right)}{\exp\left(\frac{s_{ii}^+}{\tau}\right) + \sum_{j \in \mathcal{N}_i} w_{ij} \exp\left(\frac{s_{ij}^- - m_{ij}}{\tau}\right)}, \tag{30}$$

where $s_{ii}^+$ and $s_{ij}^-$ denote the cosine similarities of positive and negative embedding pairs, respectively; $\tau$ is a temperature hyperparameter, $\mathcal{N}_i$ denotes the set of negative pairs for sample $i$. The weighting factor $w_{ij}$ and margin $m_{ij}$ are adaptive, modulated by motion length differences and inter-modality semantic similarity as detailed in Section 3.2 of the main paper. In our experiments, we set $\delta = 0.5$, $\lambda = 1.0$, and $\rho = 0.4$. The total training objective is:

$$\mathcal{L}_{\text{total}} = \mathcal{L}_{\text{recon}} + \lambda_{\text{CL}} \mathcal{L}_{\text{CL}}, \tag{31}$$

where $\lambda_{\text{CL}}$ controls the relative weighting of the contrastive objective. In our experiments, we set $\lambda_{\text{CL}} = 1.0$. This dual objective enforces that the latent space preserves input fidelity while semantically aligning the motion with other modalities.

Through these modules, MuTMoT effectively captures motion dynamics at multiple temporal scales, producing semantically expressive embeddings that integrate seamlessly into the unified multimodal embedding space.

### A.2.2 Retrieval-Augmented Latent Diffusion Model (REALM)

REALM is composed of three key components: a latent diffusion transformer, a retrieval mechanism for semantic grounding, and a temporal conditioning module designed to adaptively inject conditioning signals at each denoising timestep.

**Latent Diffusion Transformer.** Let $\mathbf{z}_0 \in \mathbb{R}^{L \times D}$ denote the initial motion latent representation, where $L$ is the sequence length in latent space and $D = 512$ is the motion embedding dimension obtained from the MuTMoT encoder. The forward diffusion process is modeled as a Markov chain:

$$q(\mathbf{z}_s \mid \mathbf{z}_{t-1}) = \mathcal{N}(\mathbf{z}_s; \sqrt{1 - \beta_s}\mathbf{z}_{s-1}, \beta_s \mathbf{I}), \tag{32}$$

$$\tag{33}$$

for $s = 1, \ldots, S$, with a linear noise schedule $\beta_s \in [10^{-4}, 0.02]$ and $S = 1000$ steps during training. The denoising model $p_\theta(\mathbf{z}_{s-1} \mid \mathbf{z}_s, \hat{\mathbf{y}}_L^{(s)})$ predicts the noise added at each step by minimizing the standard denoising score-matching objective with mean squared error:

$$\mathcal{L}_{\text{diff}} = \mathbb{E}_{\mathbf{z}_0, s, \boldsymbol{\epsilon}} \left[ \|\boldsymbol{\mu}_\theta(\mathbf{z}_s, s, \hat{\mathbf{y}}_L^{(s)}) - \boldsymbol{\epsilon}\|^2 \right], \tag{34}$$

where $\boldsymbol{\epsilon} \sim \mathcal{N}(0, \mathbf{I})$, and $\hat{\mathbf{y}}_L^{(s)}$ denotes the timestep-adaptive conditioning computed by the temporal attention mechanism. The denoising model is implemented as a 6-layer transformer with latent dimension $D = 512$, feedforward layer width of 1024, 4 attention heads, and sinusoidal timestep embeddings. Dropout with a probability of 0.1 is applied at each transformer block.

To enhance the semantic fidelity of generated motions, REALM adopts a classifier-free guidance (CFG) mechanism during inference. This technique enables the generation process to go towards more semantically meaningful outputs by interpolating between conditional and unconditional predictions.

Formally, at each denoising step $s$, two parallel forward passes are performed through the denoising network $\boldsymbol{\mu}_\theta$: (i) a conditional pass that uses the full condition context $\hat{\mathbf{y}}_L^{(s)}$, (ii) an unconditional pass with null conditioning (*i.e.*, the conditioning tokens masked out). These are computed as follows:

$$\boldsymbol{\epsilon}_{\text{cond}} = \boldsymbol{\mu}_\theta(\mathbf{z}_s, s, \hat{\mathbf{y}}_L^{(s)}), \quad \boldsymbol{\epsilon}_{\text{uncond}} = \boldsymbol{\mu}_\theta(\mathbf{z}_s, s, \emptyset). \tag{35}$$

The final noise prediction is obtained via classifier-free guidance as a linear interpolation between the two estimates:

$$\boldsymbol{\epsilon}_{\text{guided}} = \boldsymbol{\epsilon}_{\text{uncond}} + \omega \cdot (\boldsymbol{\epsilon}_{\text{cond}} - \boldsymbol{\epsilon}_{\text{uncond}}), \tag{36}$$

where $\omega = 2.75$ is the guidance scale used in our experiments determined via grid search.

**Reference Motion Retrieval and Temporal Conditioning.** For retrieval, we select the top-$k = 2$ most semantically similar motions from a large reference database using cosine similarity between conditioning embeddings (from LanguageBind) and precomputed MuTMoT motion embeddings. These retrieved sequences, along with their global latent representations, are included as reference signals for generation. To integrate conditioning information throughout the denoising process, we use a temporal conditioning module composed of learnable frame-wise tokens refined by a stack of $L_{\text{TCC}} = 3$ Time-Conditioned Cross-Attention (TCC) blocks. Each TCC block employs 4-head attention, a feedforward network of width 1024 with SquaredReLU activation, and adaptive layer normalization modulated by timestep embeddings. Dropout with probability 0.1 is applied after both attention and MLP layers. The output of the TCC stack is used to condition each step of the diffusion model, enabling temporally localized semantic grounding from both the retrieved references and the conditioning modality.

## A.3 Computational Resources and Training Configuration

All experiments were conducted using 8 NVIDIA RTX A5000 GPUs (each with 24 GB of memory) distributed across a single node. Model training and inference were implemented in PyTorch and MMCV, leveraging mixed-precision training (via PyTorch AMP) and distributed data parallelism using the NCCL backend for efficiency and scalability. Data loading was parallelized with prefetching, and all models were trained using synchronized batch normalization across GPUs.

We used the AdamW optimizer with cosine annealing and linear warmup for both MuTMoT and REALM. Gradient clipping with a maximum norm of 1.0 was applied to stabilize training. For

MuTMoT, the model was trained for 20 epochs with a batch size of 384 on each GPU, distributed across GPUs with gradient accumulation to ensure stable optimization. REALM was trained for up to 50 epochs using a 1000-step diffusion schedule during training and a reduced 50-step schedule during inference. Typical training time for MuTMoT is approximately 8 hours, while REALM training takes roughly 5 days to converge. The main training hyperparameters are summarized in Table 5.

Table 5: Training hyperparameters used for MuTMoT and REALM models.

| Parameter | MuTMoT | REALM |
|---|---|---|
| Number of GPUs | $8 \times$ RTX A5000 | $8 \times$ RTX A5000 |
| Batch Size (Each) | 384 | 160 |
| Optimizer | AdamW | AdamW |
| Learning Rate | $1 \times 10^{-4}$ | $2 \times 10^{-4}$ |
| Weight Decay | $1 \times 10^{-2}$ | $1 \times 10^{-2}$ |
| LR Schedule | Cosine w/ Warmup | Cosine w/ Warmup |
| Gradient Clipping | 1.0 (global norm) | 1.0 (global norm) |
| Epochs | 20 | 50 |
| Diffusion Steps | – | 1000 (train), 50 (test) |

## B  Additional Experiments and Ablation Studies

### B.1  Evaluation Metrics

To assess the effectiveness of our approach across different tasks, we employ a comprehensive set of evaluation metrics that measure semantic alignment, generation quality, motion reconstruction fidelity, and diversity. Below, we define the metrics used in our evaluations.

**Fréchet Inception Distance (FID).** FID measures the distributional difference between generated and real motion sequences. It computes the Fréchet distance between the Gaussian distributions fitted to the features, extracted from the official pretrained motion encoders for HumanML3D and KIT-ML as in [16], of generated and ground-truth motions:

$$\text{FID} = \|\mu_g - \mu_r\|_2^2 + \text{Tr}\left(\Sigma_g + \Sigma_r - 2(\Sigma_g \Sigma_r)^{1/2}\right), \tag{37}$$

where $(\mu_g, \Sigma_g)$ and $(\mu_r, \Sigma_r)$ denote the means and covariances of the generated and real motion features, respectively. Lower values indicate closer alignment between generated and real distributions.

**Mean Per Joint Position Error (MPJPE).** MPJPE evaluates the reconstruction accuracy of the predicted motion by computing the average L2 distance between predicted and ground-truth 3D joint positions over all frames and joints:

$$\text{MPJPE} = \frac{1}{T \times J} \sum_{t=1}^{T} \sum_{j=1}^{J} \left\| \mathbf{x}_{t,j}^{\text{pred}} - \mathbf{x}_{t,j}^{\text{gt}} \right\|_2, \tag{38}$$

where $T$ is the sequence length and $J$ is the number of joints.

**Acceleration Consistency Loss (ACCL).** ACCL evaluates temporal smoothness by computing the mean squared difference between predicted and ground-truth accelerations:

$$\text{ACCL} = \frac{1}{T - 2} \sum_{t=2}^{T-1} \left\| (\mathbf{x}_{t+1} - 2\mathbf{x}_t + \mathbf{x}_{t-1})^{\text{pred}} - (\mathbf{x}_{t+1} - 2\mathbf{x}_t + \mathbf{x}_{t-1})^{\text{gt}} \right\|_2. \tag{39}$$

**R-Precision (Top-K).** R-Precision evaluates text-motion alignment. Given one motion sample and $N$ text candidates (one ground-truth and $N-1$ negatives), it computes whether the ground-truth text appears in the top-$K$ closest matches:

$$\text{R@K} = \frac{1}{N} \sum_{i=1}^{N} \mathbb{I}\left[\text{GT}_i \in \text{Top-K}(\text{ranked}_i)\right], \tag{40}$$

where $\mathbb{I}$ is the indicator function and $N$ is the number of queries. $N = 32$ is used in our experiments as in the standard evaluations [16]. **MedR** reports the median position of the ground-truth match in the retrieval ranking list. Lower is better, indicating the true match appears closer to the top.

**Multimodal Distance (MM-Dist).** MM-Dist quantifies the alignment between the embeddings of the conditioning modality (*e.g.*, text) and the generated motion. It is computed as:

$$\text{MM-Dist} = \frac{1}{N} \sum_{i=1}^{N} \left\| \mathbf{z}_i^{\text{text}} - \mathbf{z}_i^{\text{motion}} \right\|_2, \tag{41}$$

where $\mathbf{z}_i^{\text{text}}$ and $\mathbf{z}_i^{\text{motion}}$ are the embeddings of the text and the generated motion extracted from the pretrained encoders in [16].

**Multimodality (MModality).** MModality evaluates the intra-conditioning diversity of generated motions, i.e., how much variation exists among motions generated from the same input (*e.g.*textual description). In our text-to-motion setting, we randomly sample $C$ conditioning texts, and for each text, we randomly sample two sets of $I$ motion sequences. Let $\{\mathbf{z}_{c,1}^{\text{motion}}, \ldots, \mathbf{z}_{c,I}^{\text{motion}}\}$ and $\{\mathbf{z'}_{c,1}^{\text{motion}}, \ldots, \mathbf{z'}_{c,I}^{\text{motion}}\}$ denote the extracted embeddings from the two sets conditioned on the $c$-th text. MModality is computed as:

$$\text{MModality} = \frac{1}{C \times I} \sum_{c=1}^{C} \sum_{i=1}^{I} \left\| \mathbf{z}_{c,i}^{\text{motion}} - \mathbf{z'}_{c,i}^{\text{motion}} \right\|_2, \tag{42}$$

The embeddings are extracted using the pretrained encoders from [16]. Higher values indicate greater diversity among outputs conditioned on the same input.

## B.2 Ablation Studies

To evaluate the effectiveness of the proposed architectural choices and training objectives, we conduct a series of ablation studies on both components of our framework: MuTMoT and REALM.

To better understand the contribution of individual components in the MuTMoT architecture, we conduct a comprehensive set of ablation studies. We evaluate five key aspects: (i) the use of lightweight adapter modules, (ii) dataset-specific FiLM layers, (iii) soft-gated multi-scale fusion for motion encoding, (iv) variants of contrastive loss, and (v) the impact of augmenting textual annotations through paraphrasing. All experiments are performed on the HumanML3D dataset using standard cross-modal retrieval evaluation protocols. Performance is reported in terms of Recall@K and Median Rank (MedR) for both text-to-motion and motion-to-text directions.

**Impact of Textual Paraphrasing.** To improve the diversity and generalization of text-based alignment, our final model augments each original motion caption with five paraphrased variants generated using GPT-4o. Removing this augmentation leads to a notable performance drop, particularly in Recall@1 and Recall@2, indicating that paraphrased descriptions enrich the semantic representation and improve retrieval robustness.

**Adapter vs. No Adapter.** Removing both the motion and modality-side adapter modules ($f_{\text{HM}}$, $f_{\text{MM}}$) significantly reduces retrieval performance. This suggests that the adapters are essential in aligning heterogeneous embeddings across modalities while preserving modality-specific semantics.

**FiLM Modulation.** The dataset-specific FiLM layers play a critical role in adapting the model to the distributional shifts among different datasets. Ablating FiLM results in a consistent drop in retrieval scores, validating its importance in generalizing to multi-dataset training setups.

**Multi-Scale Fusion Strategy.** To capture motion dynamics across varying temporal scales, MuTMoT aggregates the motion embedding tokens across layers via a softmax-weighted sum. When replaced with a simple use of the final layer's embedding token, the performance decreases, confirming the benefit of hierarchical temporal fusion.

**Contrastive Loss Variants.** We also evaluate the effectiveness of our contrastive learning design. While the standard InfoNCE objective provides a strong baseline, we observe measurable improvements by incorporating length-aware weighting and a dynamic semantic margin. The combination of both yields the best results, demonstrating that these enhancements improve fine-grained semantic alignment between motion and other modalities.

**Table 6:** Ablation study on MuTMoT design variants using text-to-motion and motion-to-text retrieval on HumanML3D test set. Results are reported as Recall@K and Median Rank. All results are computed on the test split.

| Model Variant | Input-to-Motion Retrieval | | | | | Motion-to-Input Retrieval | | | | |
|---|---|---|---|---|---|---|---|---|---|---|
| | R@1↑ | R@2↑ | R@3↑ | R@5↑ | MedR↓ | R@1↑ | R@2↑ | R@3↑ | R@5↑ | MedR↓ |
| Full MuTMoT | 69.56 | 85.00 | 90.45 | 94.86 | 1.00 | 70.65 | 86.92 | 91.42 | 96.26 | 1.00 |
| w/o Paraphrasing | 62.74 | 79.82 | 86.42 | 92.10 | 1.04 | 63.25 | 79.56 | 86.57 | 92.28 | 1.05 |
| w/o Adapters | 62.03 | 77.67 | 84.92 | 91.91 | 1.08 | 60.46 | 76.27 | 83.50 | 90.02 | 1.11 |
| w/o FiLM | 67.29 | 83.34 | 89.17 | 92.88 | 1.01 | 68.05 | 84.46 | 89.25 | 94.02 | 1.02 |
| Last Embedding Only | 66.07 | 81.45 | 87.95 | 93.33 | 1.02 | 61.75 | 78.92 | 86.27 | 92.96 | 1.07 |

**Table 7:** Ablation study on contrastive loss variants using text-to-motion and motion-to-text retrieval on HumanML3D test set. Results are reported as Recall@K and Median Rank. All results are computed on the test split.

| Contrastive Loss Type | Text-to-Motion Retrieval | | | | | Motion-to-Text Retrieval | | | | |
|---|---|---|---|---|---|---|---|---|---|---|
| | R@1↑ | R@2↑ | R@3↑ | R@5↑ | MedR↓ | R@1↑ | R@2↑ | R@3↑ | R@5↑ | MedR↓ |
| InfoNCE | 67.23 | 82.96 | 88.59 | 93.75 | 1.03 | 66.18 | 82.92 | 88.56 | 94.18 | 1.03 |
| + length-aware | 68.14 | 84.29 | 89.97 | 94.78 | 1.01 | 67.92 | 84.10 | 90.64 | 95.87 | 1.01 |
| + margin only | 68.48 | 83.61 | 90.08 | 94.58 | 1.02 | 68.14 | 83.69 | 90.02 | 94.99 | 1.01 |
| Full MuTMoT | 69.56 | 85.00 | 90.45 | 94.86 | 1.00 | 70.65 | 86.92 | 91.42 | 96.26 | 1.00 |

Tables 6 and 7 summarize the retrieval performance across all ablation settings. The consistent gap between the full MuTMoT model and its ablated variants highlights the complementary value of each architectural and training component.

### B.2.1 REALM Ablations

To assess the contribution of key components in the Retrieval-Augmented Latent Diffusion Model (REALM), we perform ablation studies on the KIT-ML dataset using the standard text-to-motion synthesis benchmarks. Specifically, we evaluate the effects of (i) classifier-free guidance (CFG) and temporal conditioning, (ii) conditioning granularity (frame-wise vs. global token), and (iii) retrieval-based reference motion conditioning (presence and quality). The results, shown in Tables 8 and 10, are reported using R@K (top-K retrieval precision), Frechet Inception Distance (FID), Multimodal Distance (MM Dist), and Multimodality (MModality).

**Conditioning and Guidance.** We first analyze conditioning mechanisms and guidance strategies. Removing CFG slightly decreases retrieval performance (*e.g.* R@1 drops from 43.5 to 42.4), the more notable degradation is in generation quality, with FID rising from 0.264 to 0.588. This indicates that CFG not only improves alignment with text prompts but also stabilizes generation and enhances realism. Next, we compare Frame-Wise Conditioning (FWC) with a Global Token Conditioning (GTC). FWC consistently improves all key metrics demonstrating that temporally aligned, fine-grained conditioning better preserves motion dynamics than a single global token.

**Retrieval and Textual Conditioning.** We then evaluate the contribution of semantic and structural conditioning. The inclusion of a reference motion retrieved from a large-scale motion database is designed to provide semantically relevant structure during denoising. Removing this reference signal leads to slight degradation in performance across all retrieval and diversity metrics. R@1 drops from 43.5 to 41.9, and MM-Dist slightly increases, confirming the role of retrieval in improving semantic grounding and synthesis consistency.

To evaluate the importance of explicit textual conditioning, we ablate the text encoder and condition generation solely on the retrieved reference motion along with its text encoding. This results in a substantial performance drop across all metrics: R@1 falls to 39.0, FID increases to 0.478, and MM-Dist rises sharply to 3.065. This highlights that while retrieval helps guide synthesis, semantic alignment to the text is essential for meaningful generation.

**Reference Motion Quality.** Finally, we evaluate REALM's sensitivity to reference motion quality by varying the retrieval pool size (top-$k$). Table 10 shows gradual performance degradation as less relevant references are included, yet REALM remains robust even without any reference motion (No

Ref: R@1 = 0.419, FID = 0.307). This demonstrates the model's robustness and the complementary benefit of retrieval when available.

Together, these results show that frame-wise conditioning and classifier-free guidance jointly improve realism and alignment, while retrieval conditioning contribute complementary roles in improving generation fidelity and semantic alignment, and that textual input remains a critical modality for grounding generated motion as expected.

**Table 8:** Text-to-motion generation evaluation on the KIT-ML test set. $\pm$ indicates a 95% confidence interval.

| Method | R@1 ↑ | R@2 ↑ | R@3 ↑ | FID ↓ | MM Dist ↓ | MModality ↑ |
|---|---|---|---|---|---|---|
| REALM | $0.435^{\pm.009}$ | $0.661^{\pm.010}$ | $0.785^{\pm.008}$ | $0.264^{\pm.010}$ | $2.718^{\pm.029}$ | $0.881^{\pm.079}$ |
| w/o CFG | $0.424^{\pm.011}$ | $0.648^{\pm.010}$ | $0.773^{\pm.008}$ | $0.588^{\pm.030}$ | $2.770^{\pm.047}$ | $0.895^{\pm.121}$ |
| w/o Reference Motion | $0.419^{\pm.009}$ | $0.655^{\pm.015}$ | $0.786^{\pm.014}$ | $0.307^{\pm.011}$ | $2.707^{\pm.023}$ | $0.876^{\pm.023}$ |
| w/o Text Condition | $0.390^{\pm.007}$ | $0.622^{\pm.016}$ | $0.748^{\pm.012}$ | $0.478^{\pm.026}$ | $3.065^{\pm.046}$ | $1.261^{\pm.193}$ |

**Table 9:** Frame-Wise Conditioning (FWC) vs. Global Token Conditioning (GTC) on KIT-ML.

| Method | R@1 ↑ | R@2 ↑ | R@3 ↑ | FID ↓ | MM Dist ↓ | MModality ↑ |
|---|---|---|---|---|---|---|
| REALM (w/ FWC) | $0.435^{\pm.009}$ | $0.661^{\pm.010}$ | $0.785^{\pm.008}$ | $0.264^{\pm.010}$ | $2.718^{\pm.029}$ | $0.881^{\pm.079}$ |
| REALM (w/ GTC) | $0.405^{\pm.004}$ | $0.626^{\pm.010}$ | $0.744^{\pm.005}$ | $0.325^{\pm.025}$ | $2.995^{\pm.030}$ | $0.986^{\pm.213}$ |

**Table 10:** Impact of reference motion quality (retrieval pool size, top-$k$) on KIT-ML.

| Top-$k$ | R@1 ↑ | R@2 ↑ | R@3 ↑ | FID ↓ | MM Dist ↓ | MModality ↑ |
|---|---|---|---|---|---|---|
| 2 | $0.435^{\pm.009}$ | $0.661^{\pm.010}$ | $0.785^{\pm.008}$ | $0.264^{\pm.010}$ | $2.718^{\pm.029}$ | $0.881^{\pm.079}$ |
| 20 | $0.419^{\pm.007}$ | $0.650^{\pm.006}$ | $0.775^{\pm.008}$ | $0.282^{\pm.035}$ | $2.762^{\pm.038}$ | $0.965^{\pm.220}$ |
| 50 | $0.411^{\pm.012}$ | $0.645^{\pm.017}$ | $0.773^{\pm.006}$ | $0.286^{\pm.018}$ | $2.762^{\pm.035}$ | $0.940^{\pm.286}$ |
| 100 | $0.410^{\pm.012}$ | $0.639^{\pm.005}$ | $0.772^{\pm.004}$ | $0.291^{\pm.011}$ | $2.771^{\pm.035}$ | $1.080^{\pm.124}$ |
| No Ref | $0.419^{\pm.009}$ | $0.655^{\pm.015}$ | $0.786^{\pm.014}$ | $0.307^{\pm.011}$ | $2.707^{\pm.023}$ | $0.876^{\pm.023}$ |

## B.3 Qualitative Results

### B.3.1 Comparison on HumanML3D.

Figure 5 presents qualitative comparisons of the results of text-to-motion generation on HumanML3D for three challenging prompts (as in [23]). Compared to the state-of-the-art methods such as Re-MoDiffuse, T2M-GPT, and MotionGPT, REALM generates motions that more accurately follow the describe sequences of actions and maintain spatial and temporal consistency.

### B.3.2 Any-to-Motion Generation Results

In Figure 3, we present qualitative examples that highlights the generalization capability of our model by showcasing motion generation from diverse conditioning modalities, including images and videos, despite the model being trained only on text-motion pairs. This supports the effectiveness of our unified multimodal representation.

### B.3.3 Failure Case Analysis

While REALM achieves high-quality and semantically aligned motion synthesis, certain artifacts remain, reflecting broader challenges in generative motion modeling. Figure 4 illustrates representative examples. We categorize the most common failure modes as follows:

1. **Floating Feet**: feet hover above the ground plane, indicating insufficient ground-contact constraints.
2. **Abrupt Transitions**: sudden pose or velocity changes without proper anticipation/interpolation, often at motion boundaries.

**Figure 3:** Any-to-motion generation results. Each column shows an example. Top row: input modality (image/video/audio). Bottom row: corresponding generated motion.

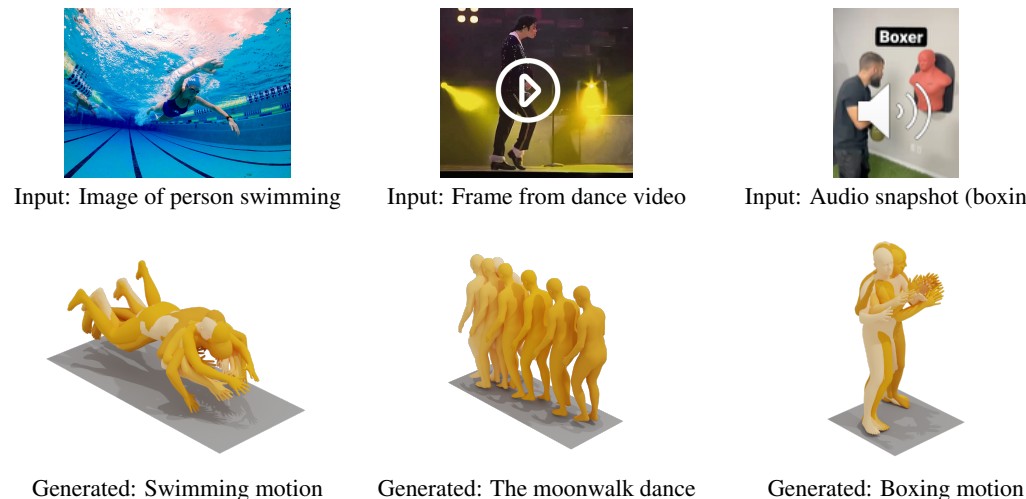

Input: Image of person swimming    Input: Frame from dance video    Input: Audio snapshot (boxing)

Generated: Swimming motion    Generated: The moonwalk dance    Generated: Boxing motion

**Figure 4: Failure case examples.** (a) Feet hover above the ground; (b) abrupt pose/velocity changes between segments; (c) high-frequency joint oscillations; (d) unintended foot sliding; (e) rotations beyond biomechanical limits; (f) missing object contact or grasp.

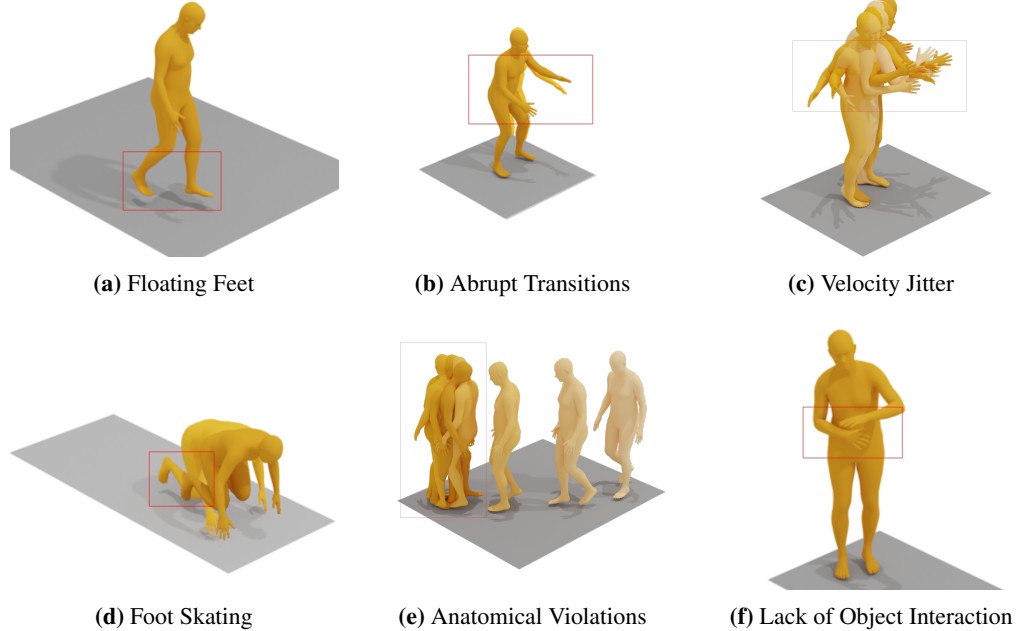

**(a)** Floating Feet    **(b)** Abrupt Transitions    **(c)** Velocity Jitter

**(d)** Foot Skating    **(e)** Anatomical Violations    **(f)** Lack of Object Interaction

3. **Velocity Jitter**: high-frequency oscillations in distal joints (e.g., wrists, ankles).

4. **Foot Skating**: unintended sliding of feet during phases that should remain stationary.

5. **Anatomical Violations**: rotations beyond biomechanical limits, producing implausible poses.

6. **Lack of Object Interaction**: motions do not convincingly reflect interactions with implied/visible objects (e.g., grasp/contact).

These artifacts are not unique to REALM, but are common across all motion synthesis models, stemming from the inherent difficulty of jointly modeling physical realism, temporal coherence, and semantic fidelity in unconstrained human motion.

**Figure 5:** Comparison of text-to-motion generation results on HumanML3D. Red phrases highlight difficult parts of the prompt.

