# OpenReview forum: "MotionBind: Multi-Modal Human Motion Alignment for Retrieval, Recognition, and Generation"
_NeurIPS.cc/2025/Conference — NeurIPS 2025 poster_

### Official Review · Reviewer_YpBN · 2025-06-28

**Clarity:** 2
**Significance:** 3
**Originality:** 3
**Rating:** 5
**Confidence:** 4

**Summary:**

This paper introduces a novel approach to multi-modal human motion alignment by integrating human motion data with existing modalities like text, vision, and audio within a unified embedding space. The authors build on the LanguageBind framework to extend its capabilities to human motion, and design the Multi-Scale Temporal Motion Transformer (MuTMoT) to encode human motion sequences into semantically meaningful embeddings, and also a retrieval-augmented latent diffusion model (REALM) for motion synthesis.

The paper demonstrates that the proposed method outperforms existing models in tasks like motion reconstruction, cross-modal retrieval, zero-shot action recognition across multiple benchmark datasets.

**Questions:**

1. In line 176 it says “This modular design ensures generalizability...” But I think the use of dataset-specific embedding modules and heads seems to limit generalizability instead of improving it. If we want to apply this method to a new, unseen dataset, wouldn’t that require re-training the embeddings and heads? I think that good generalizability means being able to generalize to a new dataset without having to re-train anything. Could the authors clarify more specifically on the generalizability of this method?

2. In the abstract (line 8-9) it says: “...Multi-Scale Temporal Motion Transformer (MuTMoT), a transformer-based encoder that...”, but in the introduction (line 49) it says: “...Multi-Scale Temporal Motion Transformer (MuTMoT), a transformer-based encoder-decoder architecture designed to...” So, is it an encoder or an encoder-decoder? The authors should make the statements consistent.

3. What does the \mathbf{w} in Eq. (3) represent? How is it obtained?

4. Are the MuTMoT part and the REALM part trained at the same time or separately?

**Ethical Concerns:**

["NO or VERY MINOR ethics concerns only"]

**Final Justification:**

Thanks to the authors for addressing my concerns, I've decided to raise my score to 5 (accept)

**Limitations:**

yes

**Paper Formatting Concerns:**

There are no major formatting issues. The paper adheres to the NeurIPS formatting guidelines. The figures are well-integrated, and the text is organized logically.

**Quality:**

3

**Strengths And Weaknesses:**

**Strengths**:

1. The integration of human motion into a multi-modal framework, allowing joint reasoning across text, vision, audio, and motion, is a unique contribution to the field.

2. The use of MuTMoT, a multi-scale transformer architecture, to capture motion dynamics at different temporal scales is a powerful method for encoding complex motion sequences. This, combined with the retrieval-augmented latent diffusion model (REALM), demonstrates the effectiveness of the approach.

3. The paper evaluates the proposed methods across a diverse set of tasks and datasets, demonstrating the robustness of the model. The paper shows competitive performance across various benchmarks.

4. The paper provides detailed descriptions of the model architectures and their components, including the MuTMoT encoder-decoder and the REALM generation model.

**Weaknesses**:

1. On text-to-motion generation on HumanML3D, the method does not deliver the best results in any of the metrics.

2. The qualitative results in the supplementary PDF only present the results of the proposed method. A qualitative comparison with other methods should be provided in order to demonstrate the strength of the proposed method.

3. While the model supports any-to-motion generation, the authors only quantitatively evaluate text-conditioned motion generation. A more comprehensive evaluation across all modalities (such as audio or video) would provide a more holistic assessment of the system's capabilities.

4. The approach assumes that the encoders from LanguageBind are sufficiently expressive for all modalities, including human motion. If LanguageBind's representations do not fully capture the nuances of motion, it might hinder the effectiveness of motion retrieval and generation.

5. The retrieval-augmented component of REALM depends on retrieving relevant motion sequences from a database. This could potentially be a bottleneck if the database is insufficient or out-of-distribution, leading to sub-optimal results.

6. While the paper is generally clear, certain sections, particularly the mathematical formulations and model descriptions, could be simplified for better accessibility.

7. The use of dataset-specific embeddings and heads seem to limit the generalizability instead of “ensuring” it, as the authors claim in line 176.

---

> ### Author Rebuttal · Authors · 2025-07-30
>
> We thank Reviewer YpBN for their thoughtful and constructive feedback. We greatly appreciate the recognition of our paper’s novelty and unique contributions in integrating human motion into a unified multimodal framework, the effectiveness and expressive power of the MuTMoT and REALM architectures, and the comprehensive evaluation and robust performance across tasks and datasets.
>
> Below, we address all the raised concerns and clarify key aspects of our design. We believe the issues noted are either due to missing clarifications or minor omissions and do not outweigh the demonstrated novelty, breadth, and rigor of our proposed framework. We respectfully hope that the reviewer will reconsider their rating, taking into account our clarifications and the demonstrated strengths, novelty, and impact of our unified multi-modal motion framework.
>
> **Response to Major Concerns:**
>
> > On text-to-motion generation on HumanML3D, the method does not deliver the best results in any of the metrics.
>    * Our method consistently delivers highly competitive performance on text-to-motion generation for HumanML3D, frequently ranking second-best, and achieves state-of-the-art results on KIT-ML as well as six additional benchmarks spanning reconstruction, cross-modal retrieval, and zero-shot action recognition. **Our primary objective is to demonstrate the effectiveness and generalizability of our multi-modal framework across a diverse set of tasks, rather than focusing exclusively on optimizing for a single dataset or metric.**
>
> > While the model supports any-to-motion generation, the authors only quantitatively evaluate text-conditioned motion generation.
>    * As acknowledged in the limitations section, our quantitative evaluation for motion generation focuses on the widely studied text-to-motion generation task, and this decision reflects the current state of the field, where standardized and robust benchmarks exist primarily for text-conditioned motion generation. While we recognize the importance of evaluating other modalities, there is currently a lack of well-established benchmarks for image- or video-conditioned motion generation beyond pose estimation. **That said, our framework is "an original step," as noted by Reviewer cqJT, to support any-to-motion generation at inference time without additional training, thanks to its multi-modal alignment.**
>    * Thus, we argue that our work should not be viewed as limited in scope. **In addition to achieving state-of-the-art or highly competitive results on text-to-motion generation, we demonstrate strong performance on six additional benchmarks across motion reconstruction, cross-modal retrieval, and action recognition.** Our primary goal is to showcase the versatility and generalizability of our multi-modal framework across diverse tasks. We believe the breadth and rigor of our evaluation present a compelling case for the effectiveness of our approach. We agree that expanding quantitative evaluation to additional conditioning modalities is a valuable direction for future work, and we plan to explore this in subsequent research.
>
> > The approach assumes encoders from LanguageBind are sufficiently expressive for all modalities, including human motion. If LanguageBind's representations do not fully capture the nuances of motion, it might hinder the effectiveness of motion retrieval and generation.
>    * We would like to clarify that LanguageBind provides a strong foundation for visual, text, and audio modalities. However, **we do not assume LanguageBind is sufficiently expressive for human motion modalities**. In fact, **central components of our method are MuTMoT’s dedicated motion encoder, which is explicitly designed to capture the nuances of human motion from pose sequences, and lightweight adapters that make LanguageBind’s implicit understanding of human movement usable by aligning it with the representations from MuTMoT’s encoder**. Our empirical results, including strong performance across cross-modal retrieval, generation, and zero-shot action recognition, demonstrate the effectiveness of this design.
>
> > The retrieval-augmented component of REALM depends on retrieving relevant motion. This could potentially be a bottleneck if the database is insufficient or out-of-distribution.
>    * While REALM utilizes reference motions to enhance generation fidelity, it continues to perform strongly with less relevant reference motions and even without retrieval, as shown in the ablation below varying the retrieval set size (top-$k$).
>
> Table 1: Impact of Reference Motion Quality. We randomly sample two references from the top-$k$ most semantically similar motions in a reference database.
> | Top-K                   | R@1 ↑        | R@2 ↑        | R@3 ↑        | FID ↓         | MM Dist ↓     | MModality ↑    |
> |--------------------------|--------------|--------------|--------------|----------------|----------------|----------------|
> | 2           | 0.435 ± .009 | 0.661 ± .010 | 0.785 ± .008 | 0.264 ± .010   | 2.718 ± .029   | 0.881 ± .079   |
> | 20          | 0.419 ± .007 | 0.650 ± .006 | 0.775 ± .008 | 0.282 ± .035  | 2.762 ± .038  | 0.965 ± .220   |
> | 50          | 0.411 ± .012 | 0.645 ± .017 | 0.773 ± .006 | 0.286 ± .018  | 2.762 ± .035  | 0.940 ± .286   |
> | 100          | 0.410 ± .012 | 0.639 ± .005 | 0.772 ± .004 | 0.291 ± .011  | 2.771 ± .035  | 1.080 ± .124   |
> | No Ref    | 0.419 ± .009 | 0.655 ± .015 | 0.786 ± .014 | 0.307 ± .011  | 2.707 ± .023  | 0.876 ± .023   |
>
> **These results show that REALM performs consistently, with only gradual performance degradation as less relevant reference motions are included or even when no reference is provided, confirming the robustness of our generation pipeline.**
>
> > A qualitative comparison with other methods should be provided.
>    * We agree that qualitative comparisons are valuable. Accordingly, we have conducted qualitative comparisons against four baseline methods (T2M-GPT, MotionDiffuse, MDM, and ReMoDiffuse), where our approach demonstrates superior or comparable motion fidelity and semantic alignment. Due to format constraints, we are unable to include the visual results here, but we will include a comprehensive version in the supplementary material.
>
> > While the paper is generally clear, certain sections, particularly the mathematical formulations and model descriptions, could be simplified for better accessibility.
>    * We appreciate your suggestion and will revise the mathematical formulations and model descriptions to improve clarity in the final manuscript.
>
> **Response to Additional Questions:**
>
> > I think the use of dataset-specific embedding modules and heads seems to limit generalizability instead of improving it... Could the authors clarify the generalizability of this method?
>    * The dataset-specific embedding modules and heads **do not** limit the model to particular datasets. **Instead, they address a practical and common challenge in motion modeling: different datasets often use different representation formats, such as varying numbers of joints or feature dimensionalities** (e.g., KIT-ML uses 21 joints with 251-dimensional vectors, while HumanML3D uses 22 joints with 263-dimensional vectors). Our dataset-specific embedding modules map these diverse raw motion formats into a unified latent space, and the corresponding heads project the shared representation back to the original format as needed.
>    * This modular design is a key enabler of generalizability and robustness. First, it allows us to train a single model across multiple heterogeneous datasets, exposing it to a wide range of motion patterns, formats, and distributions. Second, by learning from various representations during training, the model is better positioned to handle unseen datasets that share a format with any of the training sets, without requiring any additional steps. This stands in contrast to models trained on a single dataset (format), which typically need additional adaptation steps and/or retraining to generalize to new datasets. **In summary, the dataset-specific modules do not limit the model’s generalization; rather, they are what make it possible to unify and scale across diverse motion data sources within a single framework.**
>
> > Is MuTMoT an encoder or an encoder-decoder?
>    * MuTMoT is an encoder-decoder architecture, and we will update the manuscript to consistently describe it as such. For clarity, only the encoder is used for retrieval and action recognition during inference, whereas both the encoder and decoder are employed for reconstruction and generation tasks. We will make this distinction explicit in the revised manuscript to avoid any confusion.
>
> > What does the \mathbf{w} in Eq. (3) represent? How is it obtained?
>    * \mathbf{w} represents learnable contribution weights used to combine motion embeddings from different temporal scales produced by the MuTMoT encoder’s hierarchical blocks.
>    * As described in LL156-162, each block in the MuTMoT encoder captures motion information at a different temporal resolution, producing a global embedding at that scale. The weights \mathbf{w}, as noted in LL163-165, are learned during training and determine the relative importance of each temporal scale when computing the final fused motion representation. This allows the model to adaptively emphasize the most relevant temporal features.
>
> > Are the MuTMoT part and the REALM part trained at the same time or separately?
>    * We follow a two-stage training process. In the first stage, we train MuTMoT, including both the encoder and decoder, using a combination of motion reconstruction and contrastive alignment objectives, enabling it to learn robust motion representations and align them with other modalities. In the second stage, we train REALM on top of the pretrained MuTMoT encoder and decoder. This two-stage approach ensures that the motion representations are well-formed and semantically aligned before being used for conditional motion generation.

---

> > ### Comment · Reviewer_YpBN · 2025-08-05
> >
> > Thank you for addressing my concerns with new experiments and analysis. I've raised my score.

---

> > > ### Author Response · Authors · 2025-08-06
> > >
> > > Thank you for your kind response and for taking the time to review our rebuttal. We truly appreciate your constructive feedback in helping to improve our work and your willingness to raise your score.

---

### Official Review · Reviewer_cqJT · 2025-07-02

**Clarity:** 3
**Significance:** 4
**Originality:** 3
**Rating:** 5
**Confidence:** 3

**Summary:**

This paper addresses a clear gap by incorporating human motion into unified multi-modal embedding spaces, which until now have largely omitted motion data. The authors extend the LanguageBind framework to include 3D human motion, creating a shared semantic space spanning text, vision, audio, and motion. This is a notable innovation beyond prior works that aligned only motion and text. The proposed Multi-Scale Temporal Motion Transformerand the training on diverse paired datasets (language–motion, video–motion, audio–motion) enable joint semantic reasoning across all four modalities. Moreover, the introduction of REALM, a retrieval-augmented latent motion diffusion model, is an original step that allows “any-to-motion” generation – going beyond conventional text-to-motion synthesis by conditioning on arbitrary modality inputs  which is a novel capability in this domain. Overall, the idea of binding human motion into a general multi-modal embedding space and leveraging it for cross-modal retrieval/generation is fresh and significant in scope.

**Questions:**

None

**Ethical Concerns:**

["NO or VERY MINOR ethics concerns only"]

**Final Justification:**

I will maintain my recommendation.

**Quality:**

3

**Strengths And Weaknesses:**

- The paper’s approach is technically sound and well-grounded. The MuTMoT architecture is a transformer-based encoder–decoder that hierarchically encodes motion sequences at multiple temporal scales and decodes them back to full motions. This design captures both fine-grained pose dynamics and high-level action semantics, and the decoder enables accurate reconstruction of motion from the learned latent space.
- The authors conduct a comprehensive set of experiments covering motion reconstruction, cross-modal retrieval, zero-shot action recognition, and text-driven motion generation, evaluated on four datasets (HumanML3D, KIT-ML, AMASS, AIST++). The results are rigorous and highlight the method’s strengths.
- The paper is written in a clear and organized manner, making it easy to understand.

---

> ### Author Rebuttal · Authors · 2025-07-30
>
> We sincerely thank Reviewer cqJT for their highly positive feedback and recognition of our paper’s contributions. We greatly appreciate Reviewer cqJT highlighting our work’s novelty and significance in incorporating human motion into unified multi-modal embedding spaces, and the originality and technical soundness of both the proposed MuTMoT and REALM architectures.
>
> We are also encouraged by the reviewer’s acknowledgment of our comprehensive and rigorous experimental evaluation, covering motion reconstruction, cross-modal retrieval, zero-shot action recognition, and text-driven motion generation, evaluated on several benchmarks (HumanML3D, KIT-ML, AMASS, AIST++, BABEL-60, and BABEL-120).
>
> This reaffirms our belief in the potential impact our work offers toward general-purpose multi-modal human motion understanding and generation. We respectfully hope that the reviewer will continue to advocate for our paper during the discussion phase.

---

### Official Review · Reviewer_S428 · 2025-07-02

**Clarity:** 3
**Significance:** 2
**Originality:** 3
**Rating:** 4
**Confidence:** 4

**Summary:**

This paper introduces a multi-modal human motion alignment framework. For motion modality, it introduces the MuTMoT module, a Transformer-based encoder that maps motion sequences into semantically rich embeddings. The paper then extends the LanguageBind embedding space to include human motion. Cross-modal alignment is achieved through contrastive loss applied to motion-text, motion-video, and motion-video-audio pairs. Furthermore, the paper proposes REALM, a retrieval-augmented latent diffusion model for motion synthesis. Experimental results demonstrate the effectiveness of the proposed approach.

**Questions:**

1. Failure case analysis is currently missing from the paper. Could the authors provide a systematic analysis or categorization of common failure modes?

2. The paper primarily uses the AIST++ dataset, which contains music-driven dance data. How does the model generalize to other audio domains, such as speech? Have the authors tested REALM on inputs from datasets like BEAT or EMAGE, which focus on co-speech gestures? If not, could the authors comment on potential domain gaps, and whether the model would need fine-tuning or architectural adaptation to handle such input?

**Ethical Concerns:**

["NO or VERY MINOR ethics concerns only"]

**Final Justification:**

My questions have been solved.

**Limitations:**

The authors addressed the limitations and potential negative societal impact of their work.

**Quality:**

3

**Strengths And Weaknesses:**

Strengths

This work proposes an effective Multi-Modal Human Motion Alignment model that projects data from different modalities into a unified embedding space, achieving SOTA performance on retrieval tasks. The proposed REALM module effectively leverages information from multiple modalities to enhance the performance of motion synthesis.

Weakness

1.	The lack of failure case analysis is somewhat concerning. Could the authors provide insights or examples of failure cases?

2.	How does REALM respond to input conditions that are unrelated to human motion, for example, an image of a blue sky or other motion-irrelevant modality data? What kind of motion sequences would it generate in such scenarios?

3.	The model is trained on the AIST++ dataset, which associates audio with motion. As far as I know, AIST++ provides dance music audio. If the conditioning input is speech audio (such as the BEATv2[1] dataset), would REALM still be able to generate plausible motion sequences?

[1] Liu, Haiyang, et al. "EMAGE: Towards unified holistic co-speech gesture generation via expressive masked audio gesture modeling." Proceedings of the IEEE/CVF Conference on Computer Vision and Pattern Recognition. 2024.

---

> ### Author Rebuttal · Authors · 2025-07-30
>
> We thank Reviewer S428 for the thoughtful and constructive comments. We appreciate that they recognize our proposed method as effective, acknowledge the state-of-the-art performance on multi-modal retrieval, and highlight the benefits of the proposed REALM module for multi-modal motion synthesis.
> Below, we address each concern and will incorporate these valuable suggestions in the final manuscript.
>
> **Response to Major Concerns:**
>
> > Failure case analysis is currently missing from the paper. Could the authors provide a systematic analysis or categorization of common failure modes?
>    * Thank you for the valuable suggestion. We fully agree that analyzing failure cases is essential for understanding the limitations of generative motion models. Due to format constraints in the rebuttal, we are unable to provide an extensive qualitative breakdown here. However, based on the video examples included in the supplementary material, we have systematically categorized the most common failure modes as follows:
>      1. *Floating Feet*: Instances where the feet hover above the ground plane, breaking physical realism. For example, in `stands_up_from_a_laying.mp4`, the character’s feet appear to hover during the standing motion, indicating insufficient ground-contact constraints, as also observed by Reviewer Nskc.
>      2. *Abrupt Transitions*: Sudden changes in pose or velocity without appropriate anticipation or interpolation. An example, although subtle, can be found in `a_person_steps_forward_while.mp4` (00:00:23 - 00:00:25), where the transition of the arms is unnaturally sharp.
>      3. *Velocity Jitter*: High-frequency trembling or shaking in joint trajectories, particularly at the extremities. This is visible in `a_man_swings_with_his_right_arm.mp4`, where the arm visibly wobbles near the peak of the swing.
>      4. *Foot Skating*: Feet slide or drift across the floor during periods when they should remain stationary. This is observed in `a_person_walks_backwards_then.mp4` (00:01:25 - 00:02:06), where the supporting foot does not stay planted.
>      5. *Anatomical Violations*: Body parts rotate or contort beyond biomechanical limits, resulting in implausible postures. For instance, in `walks_back-and-forth.mp4` (00:03:03 - 00:04:12), the legs and torso exhibit rotations that are not physically plausible for a human.
>      6. *Lack of Object Interaction*: The generated motion fails to convincingly reflect interaction with an implied object. In `a_man_picks_up_object_from_the_ground.mp4`, the hand does not make realistic contact with the imagined object or perform a grasping motion, in part due to limited modeling of fine-grained hand articulation in SMPL.
>
>    * We would like to clarify that these issues are well-known limitations in current motion generation systems and reflect broader challenges in modeling fine-grained physical realism. Such artifacts are not unique to our method but represent active areas of research and ongoing improvement across the field. We are committed to advancing these aspects in future work, and our analysis aims to transparently contextualize both the strengths and current limitations of our approach.
>    * In the final manuscript, we will provide a structured taxonomy of these failure cases, supported by qualitative examples and visual overlays. We will also present a comprehensive analysis to contextualize the model’s behavior, along with side-by-side qualitative comparisons against relevant baselines across diverse scenarios. This will further highlight both the strengths and limitations of our approach.
>
> > How does REALM respond to input conditions that are unrelated to human motion, for example, an image of a blue sky or other motion-irrelevant modality data? What kind of motion sequences would it generate in such scenarios?
>    * Thank you for the interesting question. When presented with entirely motion-irrelevant input conditions, such as an image of a blue sky, REALM tends to produce default or low-activity motions, such as standing still or casual walking. This behavior likely reflects the model’s tendency to fall back on generic motion priors in the absence of semantically meaningful cues.
>    * In contrast, for inputs that are not fully descriptive but still weakly associated with human activity (e.g., an image of a basketball), the model often generates plausible actions such as dribbling or shooting, indicating that it leverages learned semantic associations from pretraining.
>    * We agree that analyzing such cases provides important insight into the model's behavior under ambiguous or irrelevant conditions. We will include these examples and a discussion of their implications in the failure case analysis in the final supplementary material.
>
> > The paper primarily uses the AIST++ dataset, which contains music-driven dance data. How does the model generalize to other audio domains, such as speech? Have the authors tested REALM on inputs from datasets like BEAT or EMAGE, which focus on co-speech gestures? If not, could the authors comment on potential domain gaps, and whether the model would need fine-tuning or architectural adaptation to handle such input?
>    * As shown in the supplementary visualizations, REALM is able to generate plausible motion sequences even when conditioned on audio inputs outside the music domain, such as audio of someone boxing, despite being aligned primarily with the music-driven AIST++ dataset. This generalization is largely enabled by the pre-trained LanguageBind space, which captures broad audio-video and audio-text relationships from other domains as well. However, while this demonstrates some cross-domain robustness, we have not explicitly evaluated REALM on co-speech gesture datasets like BEAT or EMAGE.
>    * While REALM produces reasonable results for speech audio with strong emotional or rhythmic cues (e.g., angry speech), its outputs are currently limited to SMPL body motion, without fine-grained hand or facial keypoints, as in SMPL-X, required by datasets like BEAT v2. Consequently, there would be domain gaps, and further fine-tuning would be necessary to fully handle fine-grained co-speech gestures. We consider this a promising direction for future work.
>
> **Closing Remarks**
>
> We would like to reiterate that our work introduces an "effective alignment model" and a generative model that "effectively leverages information from multiple modalities," as you acknowledged.
> Furthermore, we respectfully highlight that:
>    * We propose "a broadly generalizable, modular and extensible architecture for multi-modal alignment" along with "notable design of learnable frame-level tokens and time-aware modulation" for motion generation, as noted by Reviewer Nskc.
>    * Our framework is a "unique contribution to the field" with "effective and powerful proposed architectures," as noted by Reviewer YpBN.
>    * Our work "addresses a clear gap by incorporating human motion into unified multi-modal embedding spaces, which until now have largely omitted motion data." "This is a notable innovation beyond prior works that aligned only motion and text," as noted by Reviewer cqJT.
>    * "Moreover, the introduction of REALM is an original step that allows any-to-motion generation -- going beyond conventional text-to-motion synthesis by conditioning on arbitrary modality inputs, which is a novel capability in this domain," as noted by Reviewer cqJT.
>
> Taken together, we argue that these reviews affirm that "the idea of binding human motion into a general multi-modal embedding space and leveraging it for cross-modal retrieval/generation is fresh and significant in scope," as summarized by Reviewer cqJT.
>
> Finally, as noted across all reviews, our framework achieves state-of-the-art or highly competitive results across multiple multi-modal motion tasks, including cross-modal retrieval, (zero-shot) action recognition, and motion synthesis, evaluated on several benchmarks (HumanML3D, KIT-ML, AMASS, AIST++, BABEL-60, and BABEL-120).
>
> We hope these clarifications and acknowledgments help convey the strength and significance of our contributions. We respectfully hope Reviewer S428 will reconsider their rating in light of our clarifications and commitments to address their suggestions in the final manuscript.

---

> > ### Comment · Reviewer_S428 · 2025-08-05
> > **Problem Solved.**
> >
> > My questions have been solved. I will update my score.

---

> > > ### Author Response · Authors · 2025-08-06
> > >
> > > Thank you for your kind response and for taking the time to review our rebuttal. We truly appreciate your constructive feedback in helping to improve our work and your willingness to update your score.

---

> > ### Comment · Reviewer_Nskc · 2025-08-06
> >
> > We thank the authors for their detailed rebuttal and the additional experiments. While we appreciate the clarifications and new evidence, some concerns remain either unresolved or only partially addressed, as outlined below:
> >
> > ﻿
> > We acknowledge that certain artifacts are common challenges in the motion generation domain. However, the qualitative videos provided in the supplementary material exhibit artifacts to varying degrees **in nearly all cases**. In contrast, methods such as MoMask and ReMoDiffuse, despite underperforming on some metrics, **appear to produce more artifact-free visualizations**. This raises concerns about a potential conclict in REALM between achieving high quantitative scores and visual quality of generated motions.
> > ﻿
> >
> > The explanation regarding the substantial performance degradation without text paraphrasing augmentation remains a point of concern. It suggests that the method may be overly reliant on this augmentation, which could limit its robustness and generalizability.

---

### Official Review · Reviewer_Nskc · 2025-07-03

**Clarity:** 3
**Significance:** 2
**Originality:** 3
**Rating:** 3
**Confidence:** 4

**Summary:**

This paper introduces a unified framework for aligning human motion sequences with multiple modalities (text, video, audio) within a shared embedding space, alongside a novel generative pipeline for motion synthesis conditioned on arbitrary inputs. The key contributions are:
a) MuTMoT: a multi-scale temporal motion Transformer that hierarchically encodes and decodes 3D motion sequences;
b) REALM: a retrieval-augmented latent diffusion model that utilizes learnable frame tokens and cross-modal conditioning to generate high-quality motion.
The model is evaluated across several tasks, including text-to-motion generation, motion retrieval, and zero-shot action recognition.

**Questions:**

1.Can you provide quantitative or user study results for motion generation from non-text modalities (e.g., video-to-motion, audio-to-motion)?
2.Could you clarify the impact of frame-wise conditioning versus simpler global token conditioning through ablation?
3.How are positive and negative samples selected?
4.How sensitive is the model to the quality or relevance of the retrieved reference motions? And how are the candidate motion embeddings collected?

**Ethical Concerns:**

["NO or VERY MINOR ethics concerns only"]

**Final Justification:**

The rebuttal with detailed analysis and further ablation helps clarify several points of our initial review.
While the quantitative results shows outstanding performance of REALM, their effect during practical implementation to generate motion (qualitative results), is still concerning.

**Limitations:**

1. The training process is resource-intensive, requiring 8× RTX A5000 GPUs and ~5 days for REALM to converge, which may limit reproducibility and accessibility.
2. During contrastive training, all non-corresponding modality-motion pairs appear to be treated as negative samples, without consideration for potential semantic similarity. This could penalize semantically related but unmatched pairs (false negatives), potentially degrading the embedding granularity and generalization ability.

**Quality:**

3

**Strengths And Weaknesses:**

Strengths:
1.The paper proposes a modular and extensible architecture that combines multi-modal alignment, contrastive learning, and latent diffusion. The use of learnable frame-level tokens and time-aware modulation is a particularly notable design choice.
2.The model achieves strong quantitative performance on text-to-motion benchmarks (e.g., HumanML3D), outperforming existing baselines in standard metrics.
3.The supplementary ablation studies are reasonably thorough, covering most core components.
4.The architecture appears broadly generalizable to other multi-modal backbones.

Weaknesses:
1.Despite claiming robust multi-modal alignment and generative capabilities, the method relies entirely on frozen, pretrained LanguageBind encoders for all non-motion modalities (text, audio, image, video). As a result, the framework lacks novel contributions toward modality-specific understanding. Moreover, only text-conditioned generation is quantitatively evaluated, while other modalities (audio, video, image) are not assessed in the main paper.
﻿2.The supplementary generation videos exhibit noticeable artifacts, such as foot sliding and physically implausible transitions (e.g., in stands_up_from_a_laying.mp4, the subject appears to float unnaturally), which undermine the claimed motion quality.
3.Although the model achieves strong retrieval performance, a significant portion of the improvement appears to stem from GPT-4o-based text paraphrasing augmentation. As shown in Sec. B.3, Table 3, removing this augmentation causes R@1 to drop from 69.56 to 62.74. This raises concerns that the architectural contributions alone may not fully account for the observed gains. Clarifying the role of this augmentation and evaluating performance without it would strengthen the claims.
4.While training and inference are briefly described in the supplement, the paper lacks a clear, step-by-step explanation or diagram of the overall motion generation pipeline, which affects both clarity and reproducibility.

---

> ### Author Rebuttal · Authors · 2025-07-30
>
> We thank Reviewer Nskc for their thoughtful feedback and recognition of the novelty, strength of our quantitative results, and generalizability of our proposed framework. We appreciate the acknowledgment of our modular design choices and thorough ablation studies. The concerns raised primarily stem from misunderstandings or points needing clarification, which we address individually below, along with revisions planned for the final manuscript. We hope that the reviewer will reconsider their rating.
>
> **Response to Major Concerns:**
>
> > The method relies on frozen LanguageBind encoders for all non-motion modalities. The framework lacks novel contributions toward modality-specific understanding.
>    * We respectfully clarify that our method does not aim to advance modality-specific encoding (e.g., of text, video, or audio). **Instead, we address the challenge that existing large-scale, multi-modal pretrained models, such as LanguageBind, contain rich implicit knowledge about human activity due to their training on large-scale datasets with content rich in human activity, but are not directly compatible with human motion representations.**
>    * Currently, no sufficiently large-scale datasets exist with detailed human-motion annotations paired explicitly with other modalities (text, audio, images, video) at the scale required to retrain or substantially enhance encoders like LanguageBind. Given this limitation, retraining these encoders from scratch or fine-tuning them for motion tasks is impractical and data-limited.
>    * Therefore, our proposed method, introducing MuTMoT with lightweight adapters, is strategically designed as the optimal approach to exploit the latent knowledge within pretrained models like LanguageBind. Specifically, MuTMoT encodes detailed human-motion sequences and aligns them effectively to LanguageBind’s latent representations in a shared embedding space via these adapters.
>    * This design choice--recognized as **"novel and unique contribution to the field", "technically sound and well-grounded," and "effective"** by other reviewers--enables us to explicitly leverage and make the implicit multi-modal understanding already embedded in LanguageBind usable for downstream motion tasks, achieving robust multi-modal alignment and strong downstream performance across a diverse range of motion-related tasks without requiring large-scale paired motion data.
>
> > Only text-conditioned generation is quantitatively evaluated.
>    * As acknowledged in the limitations section, our quantitative evaluation for motion generation focuses on the widely studied text-to-motion generation task, and this decision reflects the current state of the field, where standardized and robust benchmarks exist primarily for text-conditioned motion generation. While we recognize the importance of evaluating other modalities, there is currently a lack of well-established benchmarks for image- or video-conditioned motion generation beyond pose estimation. **That said, our framework is "an original step," as noted by Reviewer cqJT, to support any-to-motion generation at inference time without additional training, thanks to its multi-modal alignment.**
>    * Thus, we argue that our work should not be viewed as limited in scope. **In addition to achieving state-of-the-art or highly competitive results on text-to-motion generation, we demonstrate strong performance on six additional benchmarks across motion reconstruction, cross-modal retrieval, and action recognition.** Our primary goal is to showcase the versatility and generalizability of our multi-modal framework across diverse tasks. We believe the breadth and rigor of our evaluation present a compelling case for the effectiveness of our approach. We agree that expanding quantitative evaluation to additional conditioning modalities is a valuable direction for future work, and we plan to explore this in subsequent research.
>
> > The supplementary generation videos exhibit noticeable artifacts.
>    * We acknowledge that some of the generated motions exhibit artifacts. Such issues are well-known limitations in current motion generation systems and reflect broader challenges in modeling fine-grained physical realism. **These artifacts represent an active area of research and ongoing improvement across the field.** In the final manuscript, we will include a systematic analysis of common failure modes and provide a comparative assessment against existing methods to contextualize these challenges more clearly.
>
> > Although the model achieves strong retrieval performance, removing text paraphrasing causes R@1 to drop from 69.56 to 62.74, as shown in Sec. B.3, Table 3.
>    * We agree that text paraphrasing augmentation contributes to improved retrieval performance by enhancing the model’s robustness to linguistic variation. However, we emphasize that MuTMoT’s architectural design also contributes meaningfully to the overall gains. **Specifically, the removal of core architectural components, such as adapters, leads to even larger performance drops (R@1 drops to 62.03). This indicates that while data augmentation via paraphrasing improves the model’s robustness, our framework’s architectural choices are equally crucial for achieving strong retrieval performance.** We believe that both the data and the model design play complementary roles, and we have included these ablations to transparently present their respective impacts.
>
> > While training and inference are briefly described in the supplement, the paper lacks a step-by-step explanation or diagram of the overall motion generation pipeline.
>    * While Figure 1 provides an overview of the REALM architecture and the motion generation pipeline, we agree that a more detailed, step-by-step depiction would improve both clarity and reproducibility. In the final manuscript, we will revise the figure to explicitly illustrate the full end-to-end motion generation process and include a corresponding textual walkthrough to clearly explain each stage.
>
> > During contrastive training, all non-corresponding modality-motion pairs appear to be treated as negative samples, without consideration for potential semantic similarity. This could penalize semantically related but unmatched pairs (false negatives), potentially degrading the embedding granularity and generalization ability.
>    * We respectfully disagree with the assumption that all non-corresponding modality-motion pairs are treated as negative samples. **As described in LL201-208, we introduce a dynamic semantic margin component into the InfoNCE loss, which modulates the contribution of negative pairs based on the semantic similarity between their associated non-motion inputs.**
> Specifically, if two non-motion inputs are semantically dissimilar (below a threshold in the LanguageBind space), the corresponding pair is penalized more heavily. This mechanism helps mitigate the impact of false negatives and preserves embedding granularity and generalization as shown in the ablation experiment (Table 4).
>
> **Response to Additional Questions:**
>
> > Could you clarify the impact of frame-wise conditioning versus simpler global token conditioning through ablation?
>    * Thank you for the insightful question. We conducted an ablation to directly assess the impact of Frame-Wise Conditioning (FWC) versus Global Token Conditioning (GTC), and the results are summarized in the table below:
>
> Table 1: Impact of Frame-Wise Conditioning (FWC) vs. Global Token Conditioning (GTC). Evaluation on the KIT-ML test set.
> | Method                   | R@1 ↑        | R@2 ↑        | R@3 ↑        | FID ↓         | MM Dist ↓     | MModality ↑    |
> |--------------------------|--------------|--------------|--------------|----------------|----------------|----------------|
> | REALM (w/ FWC)           | 0.435 ± .009 | 0.661 ± .010 | 0.785 ± .008 | 0.264 ± .010   | 2.718 ± .029   | 0.881 ± .079   |
> | REALM (w/ GTC)           | 0.405 ± .004 | 0.626 ± .010 | 0.744 ± .005 | 0.325 ± .025  | 2.995 ± .030   | 0.986 ± .213   |
>
> These results demonstrate that FWC provides consistent gains across all key metrics. Notably, FID improves by 18.8%, and R@1 increases by 3.0 points with FWC. This can be attributed to FWC’s ability to provide fine-grained, temporally aligned conditioning signals at every timestep, as opposed to a single global token that may lose important temporal cues.
>
> > How sensitive is the model to the quality of the reference motions?
>
>    * To assess this, we conducted an ablation varying the retrieval set size (top-$k$). We randomly sample two references from the top-$k$ most semantically similar motions in a reference database (training set of the benchmarks). Similarity is computed using cosine distance between text-conditioning embeddings and MuTMoT motion embeddings.
>
> Table 2: Impact of Reference Motion Quality
> | Top-K                   | R@1 ↑        | R@2 ↑        | R@3 ↑        | FID ↓         | MM Dist ↓     | MModality ↑    |
> |--------------------------|--------------|--------------|--------------|----------------|----------------|----------------|
> | 2           | 0.435 ± .009 | 0.661 ± .010 | 0.785 ± .008 | 0.264 ± .010   | 2.718 ± .029   | 0.881 ± .079   |
> | 20          | 0.419 ± .007 | 0.650 ± .006 | 0.775 ± .008 | 0.282 ± .035  | 2.762 ± .038  | 0.965 ± .220   |
> | 50          | 0.411 ± .012 | 0.645 ± .017 | 0.773 ± .006 | 0.286 ± .018  | 2.762 ± .035  | 0.940 ± .286   |
> | 100          | 0.410 ± .012 | 0.639 ± .005 | 0.772 ± .004 | 0.291 ± .011  | 2.771 ± .035  | 1.080 ± .124   |
> | No Ref    | 0.419 ± .009 | 0.655 ± .015 | 0.786 ± .014 | 0.307 ± .011  | 2.707 ± .023  | 0.876 ± .023   |
>
> These results show that REALM performs consistently well across a range of top-$k$ values, with only gradual performance degradation as less relevant reference motions are included. Notably, even in the absence of reference motion (No Ref), the model still achieves competitive results, confirming the robustness of our generation pipeline.

---

> > ### Comment · Reviewer_Nskc · 2025-08-06
> >
> > We thank the authors for their detailed rebuttal and the additional experiments. While we appreciate the clarifications and new evidence, some concerns remain either unresolved or only partially addressed, as outlined below:
> > ﻿
> >
> > We acknowledge that certain artifacts are common challenges in the motion generation domain. However, the qualitative videos provided in the supplementary material exhibit artifacts to varying degrees **in nearly all cases**. In contrast, methods such as MoMask and ReMoDiffuse, despite underperforming on some metrics, **appear to produce more artifact-free visualizations**. This raises concerns about a potential conflict in REALM between achieving high quantitative scores and low visual quality of generated motions.
> > ﻿
> >
> > The explanation regarding the substantial performance degradation without text paraphrasing augmentation remains a point of concern. It suggests that the method may be overly reliant on this augmentation, which could limit its robustness and generalizability.

---

> > > ### Author Response · Authors · 2025-08-07
> > >
> > > We thank the reviewer for the careful consideration and follow-up. We are glad that our clarifications helped address most of the concerns and appreciate the opportunity to further elaborate on the two remaining points:
> > >
> > > > ### Visual Artifacts in Generated Motions
> > >    - We fully acknowledge that motion generation artifacts, such as foot floating, are persistent challenges. However, we would like to clarify the context behind the qualitative examples we shared, which may help explain the discrepancy the reviewer observed.
> > >    - The supplementary videos we provided include a *randomly* selected set of motion generations from two REALM models: one trained on HumanML3D, and another on KIT-ML. Notably, many of the samples with noticeable artifacts are generated by the **KIT-ML–trained model**, which is trained on a considerably smaller and less diverse dataset. KIT-ML contains only **3,911 motion sequences** and **6,278 motion–text pairs** prior to augmentation, whereas HumanML3D offers **14,616 motion sequences** and **44,970 motion–text pairs**. This discrepancy directly impacts the quality and variability of generated motions. This holds for other models when trained on KIT-ML.
> > >    - It is also important to note that **qualitative examples shown in MoMask and ReMoDiffuse** are typically from models trained on **HumanML3D**, not KIT-ML. This makes direct visual comparison to our KIT-ML samples potentially misleading. Moreover, these works often **employ fairly strong post-processing** to reduce visual artifacts. For instance, ReMoDiffuse uses **Gaussian smoothing with $\sigma = 2.5$**, which doesn't reflect the raw generation quality.
> > >    - Regarding the quantitative-qualitative tradeoff, we recognize that models like MoMask slightly outperform REALM on FID, which could indicate marginally higher visual realism in some cases. However, our model consistently delivers **highly competitive results across multiple quantitative metrics**.
> > >    - That said, we are working to reduce these artifacts. Specifically, we have introduced **regularization terms** based on the **foot contact labels**, a 4D binary vector included in the input motion feature representation that tracks ground contact at each timestep for the left/right toe and heel. We have also explored joint rotation regularization. These additions have already resulted in **substantial improvements** in both fidelity and physical plausibility. For example, on HumanML3D, the **reconstruction error (MPJPE) improved from 25.6 → 17.7**, and **FID improved from 0.031 → 0.012**. These results indicate promising progress toward mitigating the artifacts.
> > >    - We will include these updated results, along with a **systematic failure mode analysis and qualitative comparisons with other methods**, in the final version.
> > >
> > >
> > > > ### Impact of Text Paraphrasing Augmentation
> > >
> > > We would like to clarify that **paraphrasing augmentation is applied only during training** and is **intended to enhance the model’s robustness to linguistic variation**.
> > >    - While we recognize that the drop in retrieval performance without paraphrasing may appear significant. However, this should not be seen as a weakness of our model, but rather as a reflection of the **limited linguistic diversity in HumanML3D and KIT-ML**. These datasets contain relatively few caption variants per motion, which restricts generalization. **Our paraphrasing pipeline, automated using an LLM, simulates more realistic linguistic variability without requiring additional annotation effort.**  By broadening the diversity of sentence structures, this strategy reduces the model’s sensitivity to the narrow range of phrasings present in the limited training captions.
> > >
> > > Importantly, the strong retrieval performance of our model is driven not only by data augmentation, but also by the architectural design of **MuTMoT**.
> > >    - The MuTMoT encoder is designed to capture **fine-grained motion dynamics** and has demonstrated **state-of-the-art performance in motion reconstruction**, affirming its capacity to model complex pose information.
> > >    - For effective high-level semantic understanding and cross-modal alignment, we incorporate **multi-scale global token fusion** and **lightweight adapters**. As shown in Appendix Table 3, **removing these components leads to a more substantial performance drop than removing paraphrasing alone**, underscoring their critical importance.
> > >    - Given that the task in question is text-to-motion retrieval, it is natural that linguistic diversity during training influences performance. Our paraphrasing strategy broadens this diversity, making training conditions better reflect real-world queries and ultimately improve robustness at inference.
> > >
> > > In summary, **paraphrasing serves as a complementary training augmentation**, analogous to image augmentation in computer vision. **It strengthens generalization in low-resource settings but does not substitute the core contributions of our architecture.**

---

> ### Author Response · Authors · 2025-08-07
>
> **In Closing**
>
> We believe our responses, together with the new results and additional clarifications, help address the raised concerns and reaffirm the significance of our contributions. We kindly hope the reviewer will consider these in their final assessment.
>
> Thank you again for your kind response and for taking the time to review our rebuttal. We truly appreciate your constructive feedback in helping to improve our work.

---

### Note · Authors · 2025-08-15

We sincerely thank the reviewers and ACs for their constructive feedback. It has been invaluable in strengthening the manuscript through additional clarifications, new experiments, and deeper analyses.

We are delighted that there is broad agreement across the reviews on the significance and contributions of our work:
   - It "**addresses a clear gap by incorporating human motion into unified multi-modal embedding spaces, which until now have largely omitted motion data**", representing "**a notable innovation beyond prior works that aligned only motion and text**" (Reviewer cqJT).
   - It is a "**unique contribution to the field**" with "**effective and powerful proposed architectures**" (Reviewer YpBN).
   - We propose an "**effective**", "**broadly generalizable, modular, and extensible architecture for multi-modal alignment**" and a generative model with "**notable design of learnable frame-level tokens and time-aware modulation**” that "**effectively leverages information from multiple modalities**" (Reviewers Nskc and S428).
   - "**The introduction of REALM is an original step that allows any-to-motion generation--going beyond conventional text-to-motion synthesis by conditioning on arbitrary modality inputs, which is a novel capability in this domain**." (Reviewer cqJT)

As noted across all reviews, our framework achieves **state-of-the-art or highly competitive results** on motion reconstruction, cross-modal retrieval, (zero-shot) action recognition, and motion synthesis, **spanning six diverse benchmarks**.

Taken together, we believe that these affirm that "**the idea of binding human motion into a general multi-modal embedding space and leveraging it for cross-modal retrieval/generation is fresh and significant in scope**," as summarized by Reviewer cqJT.

Following the rebuttal, **we are pleased that our responses, along with the new experiments and analyses, have fully addressed the concerns of Reviewers YpBN and S428, and resolved most of the concerns raised by Reviewer Nskc**. For the two remaining points, we appreciated the opportunity to elaborate further.

**In closing**, we believe the reviews and our clarifications highlight the novelty, technical soundness, and broad applicability of our approach, as well as its strong and consistent performance across tasks and datasets. We are grateful for the constructive dialogue and respectfully hope that our responses and additional evidence will be considered favorably in the final decision.

---

### Decision · Program_Chairs · 2025-09-17

**Decision:**

Accept (poster)

**Comment:**

## Summary

This paper introduces a unified framework for aligning human motion sequences with multiple modalities (text, video, audio) within a shared embedding space, alongside a novel generative pipeline for motion synthesis conditioned on arbitrary inputs. The key contributions are: a) MuTMoT: a multi-scale temporal motion Transformer that hierarchically encodes and decodes 3D motion sequences; b) REALM: a retrieval-augmented latent diffusion model that utilizes learnable frame tokens and cross-modal conditioning to generate high-quality motion. The model is evaluated across several tasks, including text-to-motion generation, motion retrieval, and zero-shot action recognition.

## Strengths

- The integration of human motion into a multi-modal framework allows joint reasoning across text, vision, audio, and motion, and a modular and extensible architecture combining multi-modal alignment, contrastive learning, and latent diffusion.
- Uses learnable frame-level tokens and time-aware modulation.
- Outperforms existing baselines on four datasets (HumanML3D, KIT-ML, AMASS, AIST++) with a comprehensive supplementary ablation that covered most core components.
- MuTMoT architecture encodes motion sequences at multiple temporal scales and decodes them back to full motions.

## Weaknesses

- The method relies on pre-frozen LanguageBind encoders for all non-motion modalities, limiting its modality-specific understanding.
- The model's retrieval performance is largely attributed to GPT-4o-based text paraphrasing augmentation, which may not fully account for observed gains.
- The authors only quantitatively evaluate text-conditioned motion generation, which could be a bottleneck if the database is insufficient or out-of-distribution. The paper could benefit from a more comprehensive evaluation across all modalities.
- The lack of failure case analysis raises concerns.
- The model's response to input conditions unrelated to human motion is unclear.

## Decision

The paper shows a solid contribution besides some concerns from the reviewers that have been addressed during the rebuttal. Therefore, the recommendation is the acceptance by adding and clarifying the doubts from the reviewers.

## Summarize

The discussion among the reviewers and the authors were fruitful since the authors helped the reviewers to clarify some doubts and misunderstandings during the review. At the end, most of the reviewers are in favour of accepting the paper as a poster besides the most critical reviewer.